# GRAPH NEURAL NETWORK GUIDED LOCAL SEARCH FOR THE TRAVELING SALESPERSON PROBLEM

**Benjamin Hudson**
Department of Computer Science and Technology
University of Cambridge
bh511@cam.ac.uk

**Qingbiao Li**
Department of Computer Science and Technology
University of Cambridge
ql295@cam.ac.uk

**Matthew Malencia**
Department of Electrical and Systems Engineering
University of Pennsylvania
malencia@seas.upenn.edu

**Amanda Prorok**
Department of Computer Science and Technology
University of Cambridge
asp45@cam.ac.uk

## ABSTRACT

Solutions to the Traveling Salesperson Problem (TSP) have practical applications to processes in transportation, logistics, and automation, yet must be computed with minimal delay to satisfy the real-time nature of the underlying tasks. However, solving large TSP instances quickly without sacrificing solution quality remains challenging for current approximate algorithms. To close this gap, we present a hybrid data-driven approach for solving the TSP based on Graph Neural Networks (GNNs) and Guided Local Search (GLS). Our model predicts the regret of including each edge of the problem graph in the solution; GLS uses these predictions in conjunction with the original problem graph to find solutions. Our experiments demonstrate that this approach converges to optimal solutions at a faster rate than three recent learning based approaches for the TSP. Notably, we reduce the mean optimality gap on the 100-node problem set from 1.534% to 0.705%, a $2\times$ improvement. When generalizing from 20-node instances to the 100-node problem set, we reduce the optimality gap from 18.845% to 2.622%, a $7\times$ improvement.

## 1 INTRODUCTION

Sixty years ago, the route of a delivery truck would have been fixed well before the truck departed the warehouse. Today, thanks to the availability of real-time traffic information, and the option to transmit instructions to the driver to add or remove delivery locations on-the-fly, the route is no longer fixed. Nevertheless, minimizing the length or duration of the route remains an important problem. This is an instance of the Traveling Salesperson Problem (TSP), and one of a growing number of practical applications which require solving combinatorial optimization problems in real time. In these problems, there is often a cost attributed to waiting for an optimal solution or hard deadlines at which decisions must be taken. For example, the driver cannot wait for a new solution to be computed – they may miss their deliveries, or the traffic conditions may change again. There is a need for general, anytime combinatorial optimization algorithms that produce high-quality solutions under restricted computation time. This remains challenging for current approaches, as they are specialized for a specific problem (with specific assumptions and constraints), or fail to produce good solutions quickly.

**Contributions** We present a hybrid data-driven approach for approximately solving the Euclidean TSP based on Graph Neural Networks (GNNs) and Guided Local Search (GLS), which we demonstrate converges to optimal solutions more quickly than three recent learning based approaches. We provide the following contributions:

- Our GLS algorithm is guided by the *global regret* of each edge in the problem graph. This allows the algorithm to differentiate edges that are very costly to include in the solution from ones that are not so costly, whether or not they are part of the optimal solution. Thus, using regret allows us to find high-quality, rather than optimal, solutions very quickly. We are the first

to use a measure of global regret, which we define as the cost of enforcing a decision relative to the cost of an optimal solution.

- We make this computationally tractable by approximating regret with a learned representation. Our GNN-based model operates on the *line graph* of the problem graph, which allows us to build a network without node features, focusing only on edge weight.

- We present experimental results for our approach and several learning based approaches that are aligned with recent guidelines for computational testing of learning based approaches for the TSP. Our experiments emphasize the trade-off between solution quality and time.

- We reduce the mean optimality gap on the 50-node problem set from 0.268% to 0.009%, a $30\times$ times improvement, and on the 100-node problem set from 1.534% to 0.705%, a $2\times$ improvement. When generalizing from 20-node instances to the 100-node problem set, we reduce the mean optimality gap from 18.845% to 2.622%, a $7\times$ improvement.

## 2 RELATED WORK

The Operations Research (OR) community is responsible for the majority of research in algorithms for solving combinatorial optimization problems. Concorde (Applegate et al., 2006) is widely regarded as the best exact TSP solver. As an exact solver, it can guarantee the level of optimality of the solutions it finds. It uses cutting plane algorithms (Dantzig et al., 1954; Padberg & Rinaldi, 1990; Applegate et al., 2003) and branch-and-cut methods to iteratively prune away parts of search space that will not contain an optimal solution. LKH-3 (Helsgaun, 2017), and its predecessor, LKH-2 (Helsgaun, 2009), are approximate TSP and Capacitated Vehicle Routing Problem (CVRP) solvers based on the $\kappa$-opt heuristic (Lin & Kernighan, 1973). While these solvers do not provide any guarantees, experimentation has demonstrated that they are extremely effective. Concorde, LKH-2 and LKH-3 are highly specialized and efficient solvers which can solve challenging TSPs in seconds.

However, practical real-time routing problems often impose constraints on top of the basic TSP or CVRP problem definitions. Highly-specialized solvers are difficult to adapt to these new constraints. Thus, there is a need for general algorithmic frameworks that can produce high-quality solutions with minimal computation time. To address this, Arnold & Sörensen (2019a) introduce KGLS, a GLS algorithm for the CVRP that is guided by three engineered features. They demonstrate that their algorithm can find solutions almost as good as those found by state-of-the-art metaheuristics in a fraction of the time. Our work also aims to address this gap.

Machine learning offers a way to construct flexible, yet effective combinatorial optimization algorithms. This area of research has existed for more than three decades (Smith, 1999), yet new neural architectures, especially GNNs, have driven a surge of activity in this area. As noted by Bengio et al. (2021); Cappart et al. (2021), classical combinatorial optimization approaches solve each problem instance in isolation, overlooking the fact that in practice, problems and their solutions stem from related data distributions. Machine learning offers a way to exploit this observation. We classify learning based approaches for solving the TSP into the three categories identified by Bengio et al. (2021); our approach belongs to the second category, described below.

**ML alone provides a solution.** These approaches use a machine learning model to output solutions directly from the problem definition. Vinyals et al. (2015) propose PtrNet, a sequence-to-sequence model based on LSTMs, which iteratively constructs a solution by outputting the permutation of the input nodes. They train their model using expert solutions from Concorde. They use beam search (Reddy, 1977) to produce better solutions than those sampled from their model directly. Bello et al. (2016) present a method of training PtrNet without supervision using policy gradients. Their "active search" method resembles early neural network-based approaches to combinatorial optimization, in which a model was trained online to solve a particular problem instance (Smith, 1999). This method effectively solves each problem in isolation, thus it does not leverage expertise extracted from distribution of training problem instances. Kool et al. (2018) take a similar approach to Bello et al. (2016) but use a Transformer architecture (Vaswani et al., 2017) rather than an LSTM. Their model can also be seen as a Graph Attention Network (GAT) (Veličković et al., 2017) applied to a complete graph. Kwon et al. (2021) also take a similar approach but leverage the symmetry of many combinatorial optimization problems to improve performance.

Joshi et al. (2019) train a model to produce a heatmap of which edges in the TSP graph are likely to be part of the optimal solution, and reconstruct valid tours using beam search. Similar to Vinyals et al. (2015), they train their model using expert solutions from Concorde. Kool et al. (2021) extend

the work of Joshi et al. (2019) to VRPs, and use a dynamic programming method to construct tours. Also in this category, Fu et al. (2021) train a model to output a heatmap for small TSP instances to and sample small sub-graphs from large instances to construct heatmaps for large instances. They train their approach using reinforcement learning, and use Monte Carlo tree search (MCTS) to construct valid tours.

**ML provides information to an OR algorithm.** Machine learning may not be suitable to solve a combinatorial optimization problem alone. Instead, it can provide information to augment a combinatorial optimization algorithm. Deudon et al. (2018) use a model based on the Transformer architecture to construct solutions iteratively and train it using policy gradients (they take a similar approach to Kool et al., 2018). They apply a 2-opt local search heuristic to solutions sampled from their model. In contrast to prior work, we are the first to use a machine learning model to approximate *regret*. Furthermore, many previous works that produce predictions on edges (Joshi et al., 2019; Kool et al., 2021; Fu et al., 2021) construct tours using heuristics, whereas to our knowledge, we are the first to use predictions to inform a metaheuristic (GLS).

**ML makes decisions within an OR algorithm.** Finally, a machine learning model can be embedded inside a combinatorial optimisation algorithm. In this paradigm, a master algorithm controls the high-level procedure while repeatedly calling a machine learning model to make lower level decisions. Dai et al. (2017) present a model that uses a GNN to learn an embedding of the current partial solution and a DQN (Mnih et al., 2015) to iteratively select nodes to insert using a cheapest insertion heuristic. They also use an "active search" method when solving the TSP. Wu et al. (2020) and da Costa et al. (2020) present policy gradient algorithms to learn policies to apply 2-opt operations to existing feasible solutions. Wu et al. (2020) use a Transformer architecture. da Costa et al. (2020) use a combination of GCN and RNN modules, and acheive better results. These approaches can either be seen as end-to-end learning approaches belonging to the first category, or as local search procedures where an ML model decides where to apply an operator.

## 3    Preliminaries

**Traveling Salesperson Problem** We focus on the Euclidean TSP, although our approach can be applied to other routing problems, such as the CVRP. A problem with $n$ cities, typically denoted TSP$n$, is represented as a complete, undirected, weighted graph $G = (V, E)$ with $n$ nodes. The edges $E$ represent connections between cities and are weighted by the Euclidean distance between the adjacent nodes. A solution, or *tour*, is a Hamiltonian cycle: a closed path through the graph that visits every node exactly once. An optimal solution is a cycle of minimum weight.

**Regret** Regret measures the future cost of an action that also yields an immediate reward, and is typically used to make greedy combinatorial optimization algorithms less myopic. Generally, regret is computed over a fixed horizon. For example, Potvin & Rousseau (1993) evaluate the cost of inserting a node in the best versus second-best position when constructing a CVRP solution. Hassin & Keinan (2008) solve the TSP using regret, allowing a greedy construction heuristic to remove the most recently inserted node. It is not computationally feasible to calculate regret over a global horizon, for example, for all possible insertion sequences. However, if it were possible, a greedy algorithm could compute an optimal solution by selecting the lowest regret decision at each step.

**Local Search** Local Search (LS) is a general improvement heuristic. Starting from an initial solution, local search iteratively moves to *neighboring* solutions that are lower cost than the current solution according to the objective function $g(s)$. Neighboring solutions are solutions that are reachable from a given solution by applying a certain function, or *operator*. The set of all solutions reachable from another solution by applying an operator define the neighborhood of that operator. The algorithm terminates when all neighboring solutions are inferior to the current solution, meaning the search has reached a local optimum.

**Guided Local Search** Guided Local Search (GLS) is a metaheuristic that sits on top of LS and allows it to escape local optima (Voudouris & Tsang, 1996). To apply GLS, the designer must define some *aspects* of a solution. When trapped in a local optimum, the algorithm penalizes certain aspects of the current solution which are considered to be unfavorable. The underlying LS procedure searches using an objective function that is augmented by these penalties, thus it is incentivized to

remove heavily penalized aspects from the solution. The augmented objective function $h(s)$ is

$$h(s) = g(s) + \lambda \sum_{i=0}^{M} p_i I_i(s), \tag{1}$$

where $s$ is a solution, $g(s)$ is the objective function, $\lambda$ is a scaling parameter, $i$ indexes the aspects, $M$ is the number of aspects, $p_i$ is the current number of penalties assigned to aspect $i$, and $I_i$ is an indication of whether $s$ exhibits aspect $i$, i.e.

$$I_i(s) = \begin{cases} 1 & \text{if } s \text{ exhibits aspect } i, \\ 0 & \text{otherwise.} \end{cases} \tag{2}$$

For the TSP, aspects of the solution are often defined as the edges in the problem graph. Therefore, $I_i(s)$ indicates if an edge is in the solution $s$. Upon reaching a local optimum, which aspects are penalized is determined by a utility function. The utility of penalising aspect $i$, $\text{util}_i$, is defined as

$$\text{util}_i(s_*) = I_i(s_*) \frac{c_i}{1 + p_i}, \tag{3}$$

where $s_*$ is the solution at a local optimum, $I_i(s_*)$ indicates if the solution exhibits aspect $i$, and $c_i$ is the cost of the aspect $i$. The cost of an aspect measures how unfavorable it is. The higher the cost, the greater the utility of penalizing that aspect. In the context of the TSP, the cost can be the weight of the edge (Voudouris & Tsang, 1999), or a combination of various features (Arnold & Sörensen, 2019a). Conversely, the more penalties assigned to that aspect, the lower the utility of penalising it again. The aspects with the maximum utility are always penalized, which means increasing $p_i$ by one. This penalization mechanism distributes the search effort in the search space, favoring areas where a promise is shown (Voudouris & Tsang, 1996).

We use a variation of the classical GLS algorithm (see Voudouris & Tsang, 1999) that applies alternating optimisation and perturbation phases (see Arnold & Sörensen, 2019a). During an optimization phase, the local search procedure is guided by the original objective function $g$. During a perturbation phase, it is guided by the augmented objection function $h$. After an edge is penalized, the local search is applied *only* on the penalized edge. That is, only operations that would remove the penalized edge are considered. This differs from the local search during the optimization phase, which considers all solutions in the neighborhood of the given operator. The perturbation phase continues until $K$ operations (operations that improve the solution according to the augmented objective function $h$) have been applied to the current solution. These operations perturb the solution enough for the local search to escape a local minimum. The alternating phases continue until the stopping condition is met.

## 4 METHOD

Our hybrid method, shown in Figure 1, combines a machine learning model and a metaheuristic. Our GNN-based model learns an approximation of the *global regret* of including each edge of the problem graph in the solution. The metaheuristic, GLS, uses this learned regret conjunction with the original problem graph to quickly find high-quality solutions. The learned regret allows the algorithm to differentiate between edges which are costly to include in the solution and ones that are not so costly, thus improving its ability to steer the underlying local search procedure out of local minima and towards promising areas of the solution space.

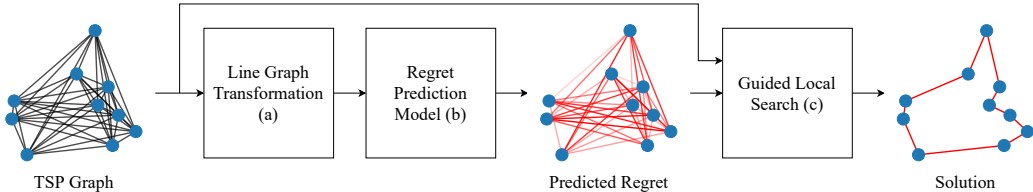

Figure 1: From a TSP formulated as a graph, we take the line graph (a) and input it into our regret approximation model (b), which predicts the regret of including each edge in the solution. GLS (c) uses these predictions in conjunction with the original problem graph to quickly find a high-quality solution.

## 4.1 GLOBAL REGRET

We define *global regret* as the cost of requiring a certain decision to be part of the solution relative to the cost of a globally optimal solution. Unlike previous heuristics using regret, which calculate the cost of a decision relative to some fixed number of alternatives (for example, the next best, or two next best options), our regret is measured relative to a *global optimal solution*. Decisions that are part of an optimal solution have zero regret, and all other decisions have positive regret. Mathematically,

$$r_i = \frac{g(s_i^*)}{g(s^*)} - 1, \tag{4}$$

where $r_i$ is the regret of decision $i$, $g$ is the objective function, $s_i^*$ is an optimal solution with $i$ fixed, and $s^*$ a globally optimal solution. With perfect information, a greedy algorithm could construct an optimal solution by sequentially selecting the lowest-regret decisions.

In the TSP, decisions correspond to which edges are included in the solution, thus regret is defined as the cost of requiring that a certain edge be part of the solution. We posit that using regret is preferable to directly classifying which edges are part of the optimal solution (which is the approach taken by Joshi et al., 2019). Where classification can produce a probability that the edge is part of *optimal* solution, regret can differentiate between edges that are very costly to have as part of the solution and ones that are not so costly, whether or not they are part of the optimal solution. Thus, using regret furthers our goal of finding high-quality, rather than optimal, solutions with minimal computation time.

## 4.2 REGRET APPROXIMATION MODEL

Calculating the global regret of an edge in the TSP graph requires solving the TSP itself, which is computationally intractable. Instead, we aim to learn a function $\hat{r}_{ij}$ that approximates the regret of an edge $r_{ij}$. We use GNNs to achieve this, as they are universal function approximators that operate on graph-structured data.

**Input transformation**  Typically, GNNs aggregate messages and store states on the nodes of a graph (Gilmer et al., 2017). Instead, we input the line graph of the original problem graph into our model. The line graph $L(G)$ of an undirected graph $G$ is a graph such that there exists a node in $L(G)$ for every edge in $G$, and that for every two edges in $G$ that share a node, there exists an edge between their corresponding nodes $L(G)$. Figure 2 illustrates this transformation for a simple graph. The result is that our model aggregates messages and stores states on the edges of the problem graph (the nodes of the line graph). Primarily, this allows us to build models with no node features, which is advantageous as the edge weights, not the specific node locations, are relevant when solving the TSP. For a complete, undirected graph $G$ with $n$ nodes, there are $n(n-1)/2$ nodes and $n(n-1)(n-2)$ edges in $L(G)$. Thus, although $G$ is complete, $L(G)$ can be very sparse.

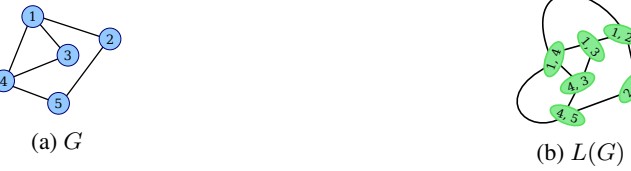

(a) $G$        (b) $L(G)$

Figure 2: An example of a graph and the corresponding line graph. The edges in $G$ are the nodes in $L(G)$, and vice-versa.

**Model architecture**  Our model consists of an embedding layer, several GNN layers, and an output layer. The embedding layer is an edge-wise fully connected layer that computes $d_h$-dimensional edge embeddings from $d_x$-dimensional edge features. Node features, if used, are concatenated onto the feature set of the adjacent edges. The forward pass of the embedding layer is written as

$$\mathbf{h}_{ij}^0 = \mathbf{W}\mathbf{x}_{ij} + \mathbf{b}, \tag{5}$$

where $\mathbf{h}_{ij}^0$ is the initial embedding of edge $ij$, $\mathbf{W}$ is a learnable weight matrix, $\mathbf{x}_{ij}$ are the input features of edge $ij$ (including any node features), and $\mathbf{b}$ is a set of learnable biases. We use $d_x = 1$ and $d_h = 128$. The edge embeddings are updated using $T$ message passing layers. Inspired by the

encoder layer of Kool et al. (2018), each layer consists of multiple sublayers. The forward pass is given by

$$\dot{\mathbf{h}}_{ij}^{t+1} = f_{\text{BN}}^t(\mathbf{h}_{ij}^t + f_{\text{MHA}}^t(\mathbf{h}_{ij}^t, L(G))), \tag{6}$$

$$\mathbf{h}_{ij}^{t+1} = f_{\text{BN}}^t(\dot{\mathbf{h}}_{ij}^{t+1} + f_{\text{FF}}^t(\dot{\mathbf{h}}_{ij}^{t+1})), \tag{7}$$

where $f_{\text{MHA}}$ is a multi-headed GAT layer (Veličković et al., 2017), $f_{\text{FF}}$ is a feedforward layer, $f_{\text{BN}}$ is batch normalisation (Ioffe & Szegedy, 2015), and $\dot{\mathbf{h}}_u^{t+1}$ is a hidden state. The layers do not share parameters. The GAT layer uses $M = 8$ heads and dimensionality $d_h/M = 16$, and the FF layer uses one hidden sublayer with dimensionality 512 and ReLU activation. Finally, the output layer is a single edge-wise fully connected layer that computes a one-dimensional output from the $d_h$-dimensional node embeddings computed by the final message passing layer. This is written as

$$\hat{r}_{ij} = \mathbf{W}\mathbf{h}_{ij}^T + \mathbf{b}, \tag{8}$$

where $\hat{r}_{ij}$ is the output for edge $ij$ and $\mathbf{h}_{ij}^T$ is the final embedding of that edge.

### 4.3 REGRET-GUIDED LOCAL SEARCH

We adapt GLS to use regret to solve the TSP, including how the initial solution is built, the local search procedure, and the perturbation strategy. Our GLS uses alternating optimization and perturbation phases. During an optimization phase, the local search procedure greedily accepts changes to the solution until it reaches a local minimum. During a perturbation phase, the algorithm penalizes and attempts to remove edges in the current solution with high regret, thus allowing it to escape local minima while simultaneously guiding it towards promising areas of the solution space, i.e., those with low regret. Effectively, the regret predictions produced by our model allow our GLS algorithm to undo costly decisions made during the greedy optimization phase.

**Initial solution**  We use a greedy nearest neighbor algorithm to construct an initial solution. Beginning from the origin node we iteratively select the lowest-regret edge leading to an unvisited node, until all nodes have been visited.

**Local Search neighborhoods**  Our LS procedure uses two solution operators for the TSP, *relocate* and *2-opt*. It alternates between using either operator, and uses a "best improvement" strategy, meaning that it exhaustively searches the neighborhood corresponding with the current operator and accepts the solution that improves the objective function the most before continuing with the other operator. The algorithm terminates when no improvement can be found in either neighborhood. The relocate operator simply changes the position of a single node in the tour. The 2-opt operator selects two nodes in the tour to swap. This divides the tour into three segments: an initial segment, an intermediate segment, and a final segment. The tour is reassembled beginning with the initial segment, the intermediate segment in reverse order, and the final segment. It is a special case of the $\kappa$-opt operator (Lin & Kernighan, 1973), although it was introduced earlier (Croes, 1958).

**Perturbation strategy**  We define the cost of an edge as the predicted regret $\hat{r}_{ij}$ of that edge. The utility of penalizing edge $ij$, $\text{util}_{ij}$, is therefore defined as

$$\text{util}_{ij}(s_*) = I_{ij}(s_*)\frac{\hat{r}_{ij}}{1 + p_{ij}}, \tag{9}$$

where we remind the reader that $s_*$ is the solution at a local optimum, $I_{ij}(s_*)$ indicates if the solution contains edge $ij$, and $p_{ij}$ is the number of penalties assigned to that edge. The edges of maximum utility are penalized. Afterward, the local search is applied *only* on the penalized edge. That is, only operations that would remove the penalized edge are considered.

## 5 EXPERIMENTS

We train our model using supervised learning. It uses a single input feature, edge weight (i.e. the Euclidean distance between adjacent nodes), to predict the regret of the edges in the problem graph. Further details of our implementation are found in Appendix A. While further input features could be considered, they come at a computational cost, as seen in Table 4. In Appendix D, we compare 11 input features, and demonstrate that edge weight is the most important feature when predicting the regret of an edge.

**Evaluation** We evaluate the trade-off between solution quality and computation time when solving randomly generated TSPs as well as TSPLIB instances[1]. Where available, we use publicly available implementations and pre-trained models. There is no publicly available implementation of Arnold & Sörensen (2019a), so we use the GLS implementation described in section 4.3 with the guides described in their paper.

We compare our approach to the following methods, the first six of which are non-learning based, and the other three are learning based:

1. Nearest Neighbour
2. Farthest Insertion
3. Local Search (as described in section 4.3)
4. Concorde (Applegate et al., 2006)
5. LKH-3 (Helsgaun, 2017)
6. *Knowledge-guided local search for the vehicle routing problem* (Arnold & Sörensen, 2019a)
7. *Attention, learn to solve routing problems!* (Kool et al., 2018)
8. *An efficient graph convolutional network technique for the travelling salesman problem* (Joshi et al., 2019)
9. *Learning 2-opt heuristics for the traveling salesman problem via deep reinforcement learning* (da Costa et al., 2020)

Following *Guidelines for the computational testing of machine learning approaches to vehicle routing problems* (Accorsi et al., 2021), we conduct two types of experiments. 1) we leave both computation time and solution quality unfixed. We measure the mean optimality gap and computation time per instance across the entire problem set. We use implementations as-described. 2) we fix computation time and measure the solution quality, in terms of mean optimality gap and the number of instances solved optimally. This allows for a *direct* comparison of approaches that was impossible in the first experiments and previous works. Where possible, we modified implementations to run continuously until the computation time limit is reached. We allow up to 10 seconds to solve each problem, including the time required to calculate input features, evaluate a model, or to construct an initial solution.

We evaluate the problem sets one instance at a time, which allows us to accurately measure computation time per instance. We allow parallelism in the GPU to sample multiple solutions to a single problem at once, where implementations are configured to do so. We use a common, single CPU and single GPU setup for all experiments. More details on the experimental setup and the hyperparameters of different approaches are given in Appendix A.

We conduct both types of experiments on three problem sets, TSP20, TSP50, and TSP100, consisting of one thousand 20, 50, and 100-node 2D Euclidean TSP instances, respectively. We generate the instances by uniform-randomly sampling node locations in the unit square $[0, 1]^2$, which is in line with the methods used by Kool et al. (2018); Joshi et al. (2019); da Costa et al. (2020). In the second type of experiment (fixed computation time), we focus on the learning based approaches and keep Concorde as a benchmark. Furthermore, we evaluate the generalization performance of the learning based approaches from smaller to larger problem sets, and from the randomly generated instances to TSPLIB instances, which are meant to represent real-world problems.

**Results** The performance of each approach when computation time and solution quality are unfixed is shown in Table 1. Approaches that are not dominated by others (i.e. they produce higher-quality solutions or are faster) are bolded. These values form a Pareto front for the trade-off between solution quality and computation time. Non-learning and learning approaches are considered separately. Our approach always lies on the Pareto front. These results are visualized in Figure 3 in Appendix B.1. The results presented for Joshi et al. (2019) differ from the results presented by the authors, due to an implementation error we found in their code (see Appendix A for the correction).

To better evaluate this trade-off, the second experiment fixes computation time and measures the solution quality as it evolves over time. Figure 4 in Appendix B.2 visualizes the performance as a function of time, which provides a rich understanding of the trade-off made by each approach. Table 2 shows the performance of the learning based approaches after 10 seconds of computation time per instance. Our approach finds better solutions on average than the other approaches for all problem sets. Notably, we reduce the mean optimality gap on the 50-node problem set from 0.268%

---

[1] Downloaded from `http://comopt.ifi.uni-heidelberg.de/software/TSPLIB95/`

| Problem | Method | Computation time (s) | Optimality gap (%) |
|---------|--------|----------------------|--------------------|
| TSP20 | Nearest Neighbor | **0.000±0.000** | **17.448±10.209** |
| | Farthest Insertion | **0.005±0.000** | **2.242±2.536** |
| | Local Search | **0.007±0.003** | **1.824±3.254** |
| | Arnold et al. | 10.009±0.008 | 0.000±0.000 |
| | Concorde | 0.119±0.083 | 0.000±0.000 |
| | LKH-3 | **0.020±0.019** | **0.000±0.000** |
| | Kool et al. | **0.038±0.005** | **0.069±0.255** |
| | Joshi et al. | 1.344±0.106 | 2.031±2.928 |
| | O. da Costa et al. | 21.120±0.352 | 0.001±0.025 |
| | Ours | **10.010±0.008** | **0.000±0.000** |
| | Kwon et al.[*] | – | 0.00 |
| | Fu et al.[*] | – | 0.000 |
| TSP50 | Nearest Neighbor | **0.002±0.000** | **23.230±7.968** |
| | Farthest Insertion | **0.065±0.001** | **7.263±3.072** |
| | Local Search | 0.101±0.031 | 3.357±2.603 |
| | Arnold et al. | 10.035±0.039 | 0.040±0.180 |
| | Concorde | **0.258±0.197** | **0.000±0.000** |
| | LKH-3 | **0.069±0.030** | **0.000±0.012** |
| | Kool et al. | **0.124±0.006** | **0.494±0.655** |
| | Joshi et al. | 3.080±0.182 | 0.884±1.700 |
| | O. da Costa et al. | 24.338±0.523 | 0.136±0.314 |
| | Ours | **10.037±0.039** | **0.009±0.069** |
| | Kwon et al.[*] | – | 0.03 |
| | Fu et al.[*] | – | 0.0145 |
| TSP100 | Nearest Neighbor | **0.010±0.000** | **25.104±6.909** |
| | Farthest Insertion | 0.444±0.010 | 12.456±2.944 |
| | Local Search | 0.727±0.165 | 4.169±2.047 |
| | Arnold et al. | 10.128±0.194 | 1.757±1.240 |
| | Concorde | **0.573±0.771** | **0.000±0.000** |
| | LKH-3 | **0.118±0.022** | **0.011±0.058** |
| | Kool et al. | **0.356±0.007** | **2.368±1.185** |
| | Joshi et al. | 6.127±0.081 | 1.880±4.097 |
| | O. da Costa et al. | 30.656±0.734 | 0.773±0.717 |
| | Ours | **10.108±0.175** | **0.698±0.801** |
| | Kwon et al.[*] | – | 0.14 |
| | Fu et al.[*] | – | 0.037 |

Table 1: Solution quality and computation time for various approaches. Means and standard deviations are reported, calculated across the entire problem set. Pareto optimal values (i.e. faster or better solutions) are bolded. Some results (*) are taken directly from papers that do not report computation time per instance, and thus cannot be compared.

to 0.009%, a 30× times improvement, and on the 100-node problem set from 1.534% to 0.705%, a 2× improvement. Joshi et al. (2019) finds more optimal solutions to the TSP100 problem set, but has worse average performance than our approach. Our approach finds more optimal solutions to the TSP20 and TSP50 problem sets than the other approaches.

We evaluate the performance of our approach and other learning based approaches when generalizing from smaller instances to larger ones, and from randomly generated instances to TSPLIB instances, which are meant to represent real-world instances. These results are summarized in Appendix C.1 and Appendix C.2 respectively. Our approach generalizes well. Notably, when generalizing from 20-node instances to the 100-node problem set, we reduce the mean optimality gap from 18.845% to 2.622%, a 7× improvement.

## 6 DISCUSSION

Our approach generalizes to larger problems and to real-world problems well, which may be due to the unique input transformation we apply to the problem graph. We input the line graph $L(G)$ of the problem graph $G$, which allows the model to aggregate messages and store states on edges

| Problem | Method | Optimality gap (%) | Optimal solutions (%) |
|---------|--------|--------------------|-----------------------|
| TSP20 | Concorde | 0.000±0.000 | 100.0000 |
| | Kool et al. | 0.069±0.255 | 84.6000 |
| | Joshi et al. | 1.000±12.492 | 97.4000 |
| | O. da Costa et al. | 0.002±0.027 | 99.0000 |
| | Ours | **0.000±0.000** | **100.0000** |
| TSP50 | Concorde | 0.000±0.000 | 100.0000 |
| | Kool et al. | 0.494±0.655 | 29.5000 |
| | Joshi et al. | 1.206±18.066 | 86.3000 |
| | O. da Costa et al. | 0.268±0.492 | 48.7000 |
| | Ours | **0.009±0.069** | **96.2000** |
| TSP100 | Concorde | 0.000±0.000 | 100.0000 |
| | Kool et al. | 1.958±1.064 | 0.3000 |
| | Joshi et al. | 1.559±4.071 | **51.6000** |
| | O. da Costa et al. | 1.534±1.098 | 1.4000 |
| | Ours | **0.705±0.807** | 20.5000 |

Table 2: Solution quality measured after after 10 seconds of computation time per instance. The mean optimality gap and standard deviation, as well as the percentage of optimally solved problems are reported. Our approach converges to better solutions at a faster rate than the other approaches.

rather than nodes. This allows us to build models without node features, which is advantageous as the edge weights, not the specific node positions, are relevant when solving the TSP. Including the node positions as features, as done by Kool et al. (2018); Joshi et al. (2019); da Costa et al. (2020) may hinder the learned policy's ability to generalize. Deudon et al. (2018) apply PCA on the node positions before inputting them into their model so they are rotation invariant, yet they do not report generalization performance. For a Euclidean TSP problem graph with $n$ nodes, $L(G)$ has $n(n-1)/2$ nodes, meaning that although $G$ is complete, $L(G)$ can be very sparse, which may help learning (Cappart et al., 2021). However, the model consumes more GPU memory than models using the problem graph $G$.

Many approaches that learn construction heuristics (for example Kool et al., 2018) treat the tour as a permutation of the input nodes. By considering the solution as a sequence, these approaches miss out on its symmetry: a TSP solution is invariant to the origin city. Outputting the correct sequence becomes increasingly difficult as the number of cities increases. This has recently been addressed by Kwon et al. (2021), who leverage the symmetry of the TSP in their approach. Our model learns the regret of including a given edge in the solution based on its features and those of its neighbors, which implicitly assumes the symmetry of a TSP solution.

We argue that our hybrid architecture is more general than others, as our global regret is defined for most routing problems. When applying our method to a new routing problem, the designer need only plug-and-play a local search procedure that is appropriate for the problem. In contrast, methods which learn heuristics cannot always be transferred to other problems. For example, 2-opt cannot be used on routing problems that include pickup and drop-off constraints (Pacheco et al., 2021). A drawback of our approach is that it relies on supervised learning. In future work, our regret approximation model could be trained end-to-end.

## 7 CONCLUSION

We present a hybrid data-driven approach for solving the TSP based on Graph Neural Networks (GNNs) and Guided Local Search (GLS). To inform GLS, we use a GNN-based model to compute fast, generalizable approximations of the *global regret* of each edge. Our model operates on the line graph of the TSP graph, which allows us to build a network that uses edge weight as the only input feature. This approach allows our algorithm to escape local minima and find high-quality solutions with minimal computation time. Finally, we present experimental results for our approach and several recent learning based approaches that emphasize the trade-off between solution quality and computation time. We demonstrate that our approach converges to optimal solutions at a faster rate than the evaluated learning based approaches.

ACKNOWLEDGEMENTS

We gratefully acknowledge the support of the European Research Council (ERC) Project 949940 (gAIa).

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

## A IMPLEMENTATION DETAILS

We conduct all computational experiments on virtual machines using a single core of an Intel Xeon E5-2650 v4 processor, 24 GB of RAM, and a single Tesla P100 GPU with 16 GB of VRAM. We evaluate the various approaches one-at-a-time, to ensure they do not compete for computational resources. When evaluating the various approaches, we record actual tours and use a common method of calculating tour cost and optimality gap, as we found that small discrepancies in cost calculations between implementations led to skewed results. We recommend this approach when comparing different approaches for solving the TSP. Based on approximately 200,000 TSP solutions, we determined that an absolute threshold of $10^{-7}$ on the value of the objective function is sufficient to separate optimal and sub-optimal solutions.

We train our model on sets of one hundred thousand 20, 50, and 100-node 2D Euclidean TSP instances. We generate all instances by uniform-randomly sampling node locations in the unit square $[0, 1]^2$. We calculate the target output, the regret of including each edge in the solution, according to eq. (4). We use Concorde to find an optimal solution and LKH-3 with 100 trials and 10 runs to solve find an optimal solution with a given edge fixed. We empirically validate that this configuration for LKH-3 is sufficient to find optimal solutions for problems with 100 nodes or less. We scale the input features and target outputs to a range of $[0, 1]$.

We train our model by minimizing the $l^2$ loss to the target output using the Adam optimizer (Kingma & Ba, 2014) and an exponentially decaying learning rate $\gamma = 10^{-3} \times (0.99)^e$, where $e$ is the epoch. We train the model for 100 epochs with early stopping. We use a training batch size $B = 32$ for the 20 and 50-node training sets, and a training batch size of $B = 15$ for the 100-node training set due to limited GPU memory. Our model uses $T = 3$ message passing layers and our GLS algorithm uses $K = 20$ perturbation moves. Our approach is implemented using Python 3.8.11, PyTorch 1.9.0 (Paszke et al., 2019), DGL 0.7.1 (Wang et al., 2019), and CUDA 10.2, and is open-source.

We use the open-source implementation of Kool et al. (2018) with the best configuration, which samples 1280 solutions in parallel. We use this implementation as-is for experiments where computation time is not fixed. We modified the implementation to continuously sample 1280 solutions until the computation time limit is reached for experiments where computation time is fixed. We use the open-source implementation of da Costa et al. (2020) with the best configuration, which uses 2000 iterations. We use this implementation as-is for experiments where computation time is not fixed. We modified the implementation to iterate until the computation time limit is reached for experiments where computation time is fixed. We used the open-source implementation of Joshi et al. (2019). We found a programming error in their beamsearch implementation that caused invalid tours to be reported. We modified their implementation to detect invalid tours and discarded them when they are produced. Our changes are shown in Listing 1. Therefore, our results may differ from those originally reported by the authors.

Listing 1: Programming error fix for Joshi et al. (2019)

```
1  diff --git a/utils/model_utils.py b/utils/model_utils.py
2  index 491b630..ad554c0 100644
3  --- a/utils/model_utils.py
4  +++ b/utils/model_utils.py
5  @@ -5,6 +5,12 @@ import torch.nn as nn
6    from utils.beamsearch import *
7    from utils.graph_utils import *
8
9  +def is_valid(tour):
10 +    nodes = set(tour)
11 +    for n in nodes:
12 +        if tour.count(n) != 1:
13 +            return False
14 +    return True
15
16   def loss_nodes(y_pred_nodes, y_nodes, node_cw):
17       """
18  @@ -143,7 +149,12 @@ def beamsearch_tour_nodes_shortest(y_pred_edges,
        x_edges_values, beam_size, batc
19             if hyp_len < shortest_lens[idx] and is_valid_tour(hyp_nodes,
                    num_nodes):
20                 shortest_tours[idx] = hyp_tours[idx]
21                 shortest_lens[idx] = hyp_len
22  -    return shortest_tours
23  +
24  +    shortest_tours_list = shortest_tours.cpu().numpy().tolist()
25  +    tour_is_valid = np.array([is_valid(tour) for tour in
        shortest_tours_list])
26  +    if not tour_is_valid.all():
27  +        print(shortest_tours, tour_is_valid)
28  +    return shortest_tours, tour_is_valid
29
30
31   def update_learning_rate(optimizer, lr):
```

# B    COMPARISON TO OTHER APPROACHES

## B.1    PERFORMANCE CHARTS

Following the guidelines in Accorsi et al. (2021), we measure the mean optimality gap and computation time per instance, while both are unfixed. Figure 3 depicts the performance of the evaluated non-learning and learning based approaches averaged across the entire 20, 50, and 100-node problem sets. The axes are scaled so the plots are easily readable. Results that fall outside the plot limits are shown with an arrow pointing towards the horizontal axis at the mean computation time. These plots clearly identify Pareto optimal approaches and dominated ones.

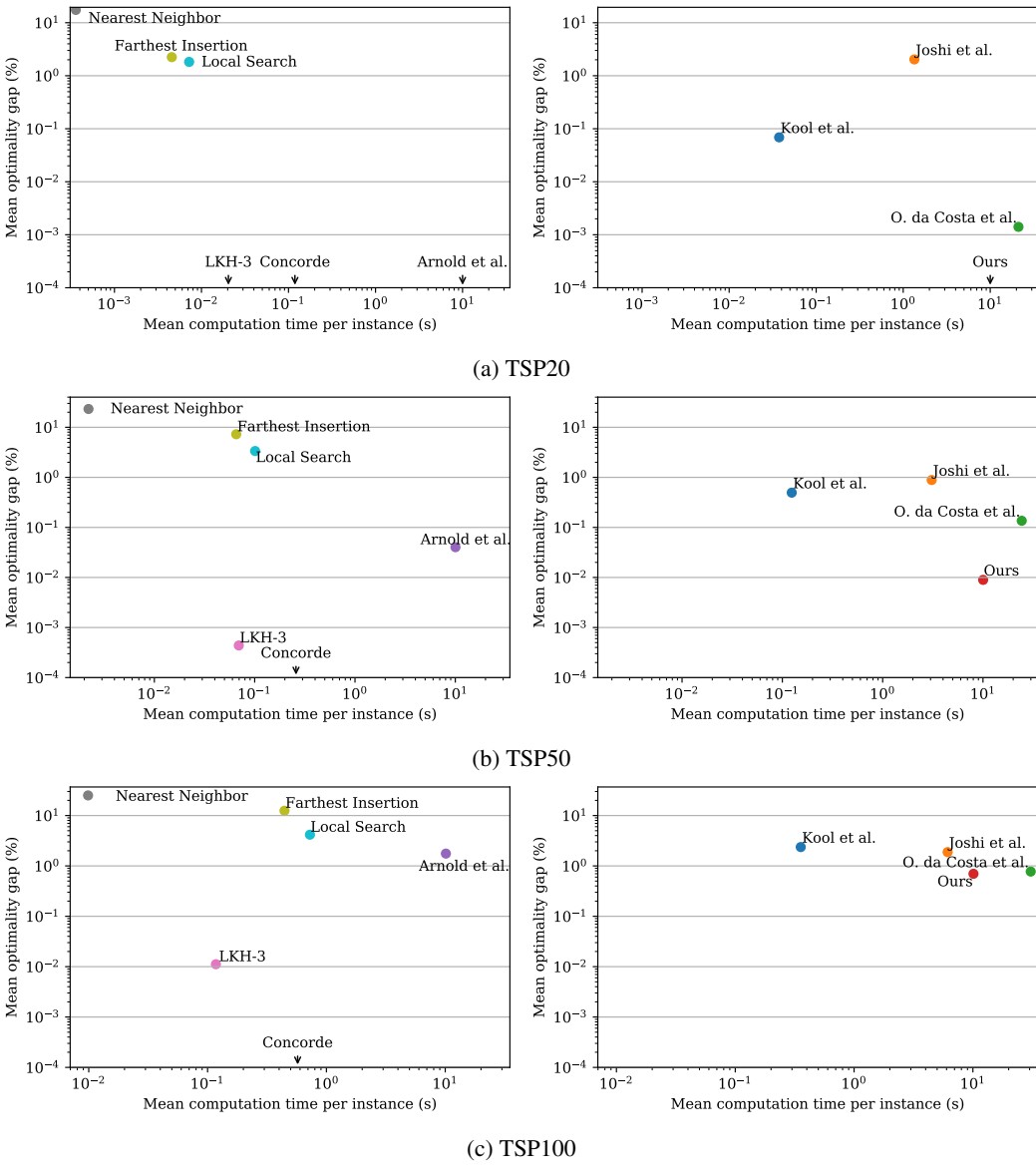

Figure 3: Mean optimality gap and computation time per instance for three increasingly difficult problem sets. The left plot shows non-learning based approaches, where the Nearest Neighbour heuristic, LKH-3, and Concorde typically form the Pareto front. The right plot shows learning based approaches, where Kool et al. (2018) and our approach form the Pareto front.

## B.2 CONVERGENCE PROFILE CHARTS

We evaluate the solution quality, in terms of mean optimality gap and number of problems solved optimally, as a function of computation time over a fixed computation time. The resulting *convergence profile* provides a detailed view of how each approach trades off between solution quality and computation time. Figure 4 depicts convergence profiles for our approach and several recent learning based approaches, for models trained and evaluated on 20, 50, and 100-node problem instances. Computation time required to compute input features, evaluate a model, or to construct an initial solution is visible as a gap between the trace and the vertical axis, and is especially noticeable for Joshi et al. (2019).

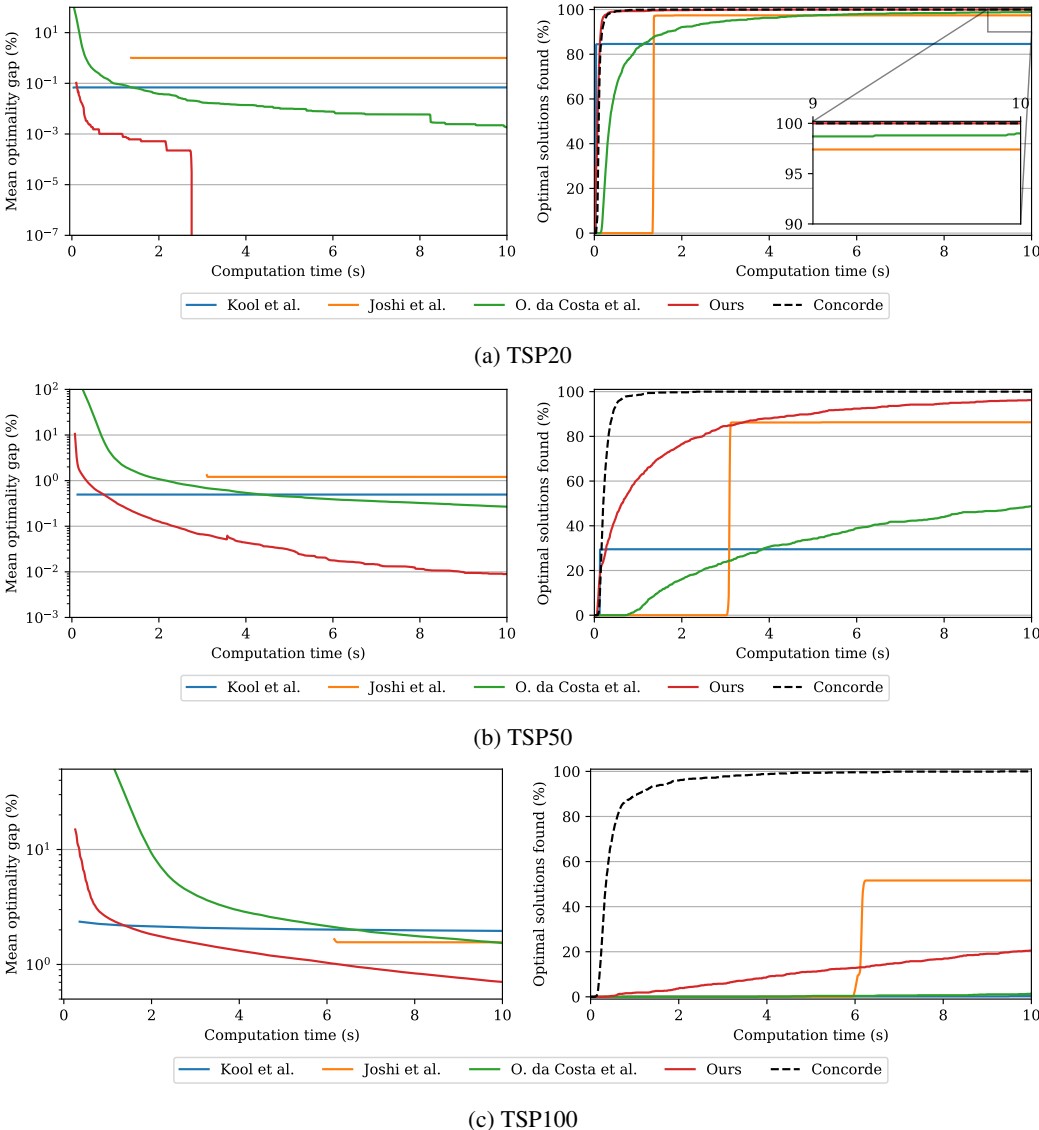

Figure 4: Solution quality as a function of computation time for three increasingly difficult problem sets, demonstrating that our method converges to optimal solutions at a faster rate than the evaluated learning based approaches. The left plot shows the mean optimality gap. The right plot shows the percentage of optimally solved problems.

# C    GENERALIZATION PERFORMANCE

## C.1    GENERALIZATION TO LARGER INSTANCES

Figure 5 depicts convergence profiles for the learning based approaches when generalizing from smaller problem instances to larger ones. The plots are arranged in order of increasing difference in size between the training and test problem sets.

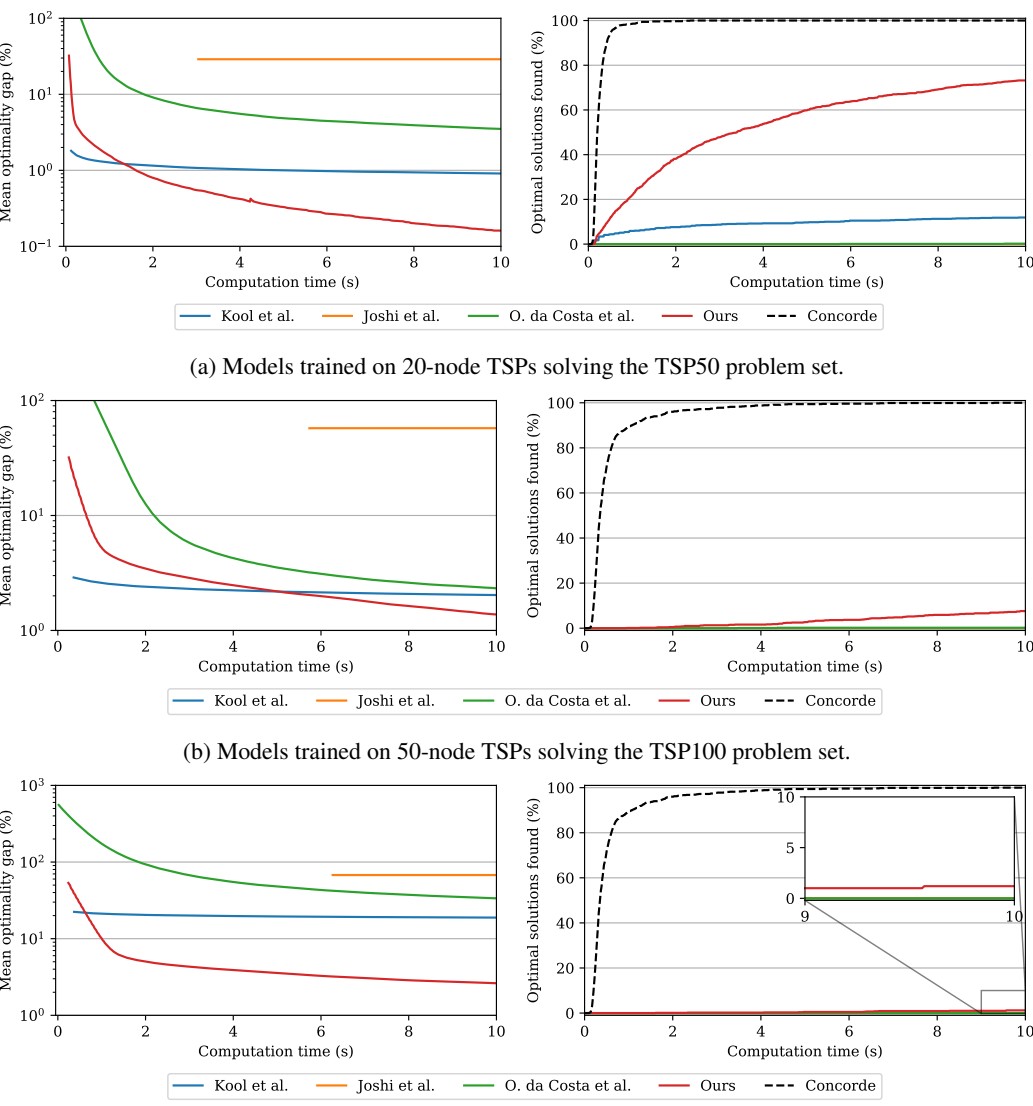

(a) Models trained on 20-node TSPs solving the TSP50 problem set.

(b) Models trained on 50-node TSPs solving the TSP100 problem set.

(c) Models trained on 20-node TSPs solving the TSP100 problem set.

Figure 5: Solution quality as a function of computation time for three increasingly difficult generalization tasks, demonstrating that our method is able to generalize from smaller to larger problem instances better than several learning based approaches. The left plot shows the mean optimality gap. The right plot shows the percentage of optimally solved problems.

## C.2 GENERALIZATION TO REAL-WORLD INSTANCES

In many cases there may be insufficient examples of real-world problem instances to train a model, meaning it must be trained on synthetic data. Therefore, it is important to evaluate this "sim-to-real" transfer. Table 3 shows mean optimality gap and computation time for the evaluated learning based methods using models trained on random 100-node TSPs and evaluated on TSPLIB instances with 50 to 200 nodes and 2D Euclidean distances (29 instances).

| Method | Kool et al. | | Joshi et al. | | O. da Costa et al. | | Ours | |
| Instance | Time (s) | Gap (%) | Time (s) | Gap (%) | Time (s) | Gap (%) | Time (s) | Gap (%) |
|---|---|---|---|---|---|---|---|---|
| eil51 | **0.125±0.000** | **1.628±0.125** | 3.026±0.012 | 8.339±0.000 | 28.051±0.040 | 0.067±0.097 | **10.074±0.068** | **0.000±0.000** |
| berlin52 | **0.129±0.001** | 4.169±1.289 | 3.068±0.022 | 33.225±0.000 | 31.874±0.388 | 0.449±0.421 | **10.103±0.048** | **0.142±0.000** |
| st70 | **0.200±0.001** | 1.737±0.093 | 4.037±0.009 | 24.785±0.000 | **23.964±0.122** | **0.040±0.085** | 10.053±0.078 | 0.764±0.000 |
| eil76 | **0.225±0.000** | 1.992±0.004 | 4.303±0.021 | 27.411±0.000 | **26.551±0.065** | 0.096±0.237 | 10.155±0.067 | **0.163±0.000** |
| pr76 | **0.226±0.002** | **0.816±0.130** | 4.378±0.023 | 27.793±0.000 | 39.485±0.152 | 1.228±0.908 | 10.049±0.023 | 0.039±0.000 |
| rat99 | **0.347±0.000** | 2.645±0.103 | 5.559±0.016 | 17.633±0.000 | 32.188±0.064 | **0.123±0.000** | 9.948±0.013 | 0.550±0.000 |
| kroA100 | **0.352±0.001** | 4.017±0.043 | 5.705±0.018 | 28.828±0.000 | 42.095±0.109 | 18.313±0.000 | 10.255±0.045 | **0.728±0.000** |
| kroB100 | **0.352±0.002** | 5.142±0.269 | 5.712±0.018 | 34.686±0.000 | 35.137±0.078 | 1.119±0.060 | **10.317±0.150** | **0.147±0.000** |
| kroC100 | **0.352±0.001** | **0.972±0.095** | 5.641±0.013 | 35.506±0.000 | **34.333±0.168** | 0.349±0.191 | 10.172±0.208 | 1.571±0.000 |
| kroD100 | **0.352±0.001** | 2.717±0.327 | 5.621±0.024 | 38.018±0.000 | 25.772±0.090 | 0.866±0.447 | **10.375±0.155** | **0.572±0.000** |
| kroE100 | **0.352±0.001** | 1.470±0.109 | 5.650±0.026 | 26.589±0.000 | 34.475±0.166 | 1.832±0.453 | **10.270±0.171** | **0.755±0.000** |
| rd100 | **0.352±0.001** | 3.407±0.092 | 5.737±0.033 | 50.432±0.000 | 28.963±0.077 | **0.003±0.005** | 10.125±0.039 | 0.459±0.000 |
| eil101 | **0.359±0.001** | 2.994±0.230 | 5.790±0.018 | 26.701±0.000 | 23.842±0.075 | 0.387±0.096 | **10.276±0.036** | **0.201±0.000** |
| lin105 | **0.380±0.001** | 1.739±0.125 | 5.938±0.040 | 34.902±0.000 | 39.517±0.145 | 1.867±0.648 | **10.330±0.037** | **0.606±0.000** |
| pr107 | **0.391±0.001** | 3.933±0.352 | 5.964±0.048 | 80.564±0.000 | 29.039±0.107 | 0.898±0.345 | **9.977±0.050** | **0.439±0.000** |
| pr124 | **0.499±0.002** | 3.677±0.523 | 7.059±0.031 | 70.146±0.000 | 29.570±0.038 | 10.322±4.603 | **10.360±0.190** | **0.755±0.000** |
| bier127 | **0.522±0.001** | 5.908±0.414 | 7.242±0.022 | 45.561±0.000 | 39.029±0.146 | 3.044±0.272 | **10.260±0.124** | **1.948±0.000** |
| ch130 | **0.550±0.001** | 3.182±0.182 | 7.351±0.074 | 39.090±0.000 | **34.436±0.160** | 0.709±0.762 | 10.032±0.063 | 3.519±0.000 |
| pr136 | **0.585±0.000** | 5.064±0.998 | 7.727±0.026 | 58.673±0.000 | **31.056±0.081** | **0.000±0.000** | 10.379±0.077 | 3.387±0.000 |
| pr144 | **0.638±0.001** | 7.641±0.516 | 8.132±0.017 | 55.837±0.000 | **28.913±0.093** | 1.526±0.276 | 10.276±0.080 | 3.581±0.000 |
| ch150 | **0.697±0.002** | 4.584±0.324 | 8.546±0.048 | 49.743±0.000 | 35.497±0.117 | 0.312±0.084 | **10.109±0.060** | 2.113±0.000 |
| kroA150 | **0.695±0.001** | 3.784±0.336 | 8.450±0.017 | 45.411±0.000 | **29.399±0.059** | 0.724±0.198 | 10.331±0.281 | 2.984±0.000 |
| kroB150 | **0.696±0.002** | 2.437±0.188 | 8.573±0.021 | 56.745±0.000 | **29.005±0.071** | 0.886±0.347 | 10.018±0.103 | 3.258±0.000 |
| pr152 | **0.708±0.001** | 7.494±0.265 | 8.632±0.035 | 33.925±0.000 | **29.003±0.106** | **0.029±0.018** | 10.267±0.096 | 3.119±0.000 |
| u159 | **0.764±0.001** | 7.551±1.581 | 9.012±0.019 | 38.338±0.000 | **28.961±0.096** | **0.054±0.000** | 10.428±0.078 | 1.020±0.000 |
| rat195 | **1.114±0.003** | 6.893±0.505 | 11.236±0.028 | 24.968±0.000 | **34.425±0.054** | 0.743±0.312 | 12.295±0.087 | **1.666±0.000** |
| d198 | **1.153±0.052** | 373.020±45.415 | **11.519±0.091** | 62.351±0.000 | 30.864±0.066 | 0.522±0.297 | 12.596±0.043 | 4.772±0.000 |
| kroA200 | **1.150±0.000** | 7.106±0.391 | 11.702±0.036 | 40.885±0.000 | **33.832±0.402** | 1.441±0.068 | 11.088±0.075 | **2.029±0.000** |
| kroB200 | **1.150±0.001** | 8.541±0.565 | 11.689±0.041 | 43.643±0.000 | **31.951±0.128** | 2.064±0.419 | 11.267±0.245 | **2.589±0.000** |
| Mean | **0.532±0.300** | 16.767±67.953 | 7.000±2.415 | 40.025±15.702 | 31.766±4.543 | 1.725±3.767 | **10.420±0.630** | **1.529±1.328** |

Table 3: Solution quality and computation time for for learning based methods using a model trained on random 100-node problems and evaluated on TSPLIB instances with 50 to 200 nodes and 2D Euclidean distances. Means and standard deviations are reported, calculated across 10 runs per problem instance. Pareto optimal values (i.e. faster or better solutions) are bolded.

# D  ANALYSIS OF ADDITIONAL INPUT FEATURES

We evaluate the potential benefit of adding additional features to our regret approximation model. While more features can help produce better predictions, they come at the cost of additional computation time. We consider a total of ten additional features, described in Table 4. We use a Gaussian process-based surrogate model to conduct global sensitivity analysis (Gratiet et al., 2016) of the input features on the validation loss of the model after training. We assume that better regret predictions will equate to better guidance by the GLS algorithm, ultimately resulting in better performance to the problem sets.

| Name | Description | Complexity |
|------|-------------|-----------|
| Node width | Perpendicular distance from a node to a line from the depot node through the centroid of the nodes (Arnold & Sörensen, 2019b). | $\Theta(n)$ |
| Edge width | Absolute difference between the width of two nodes. | $\Theta(n^2)$ |
| Depot weight | Weight from a node to the depot node. | $\Theta(n)$ |
| Neighbor rank | $k$ where node $j$ is the $k$-nearest neighbor of $i$. There are two features for a given edge, the rank of $i$ w.r.t. $j$ and the rank of $j$ w.r.t. $i$. | $\Theta(n^2)$ |
| 30%-NN graph | If the edge is part of the $k$-nearest neighbor graph, where $k$ is $0.3n$. | $\Theta(n^2)$ |
| 20%-NN graph | If the edge is part of the $k$-nearest neighbor graph, where $k$ is $0.2n$. | $\Theta(n^2)$ |
| 10%-NN graph | If the edge is part of the $k$-nearest neighbor graph, where $k$ is $0.1n$. | $\Theta(n^2)$ |
| MST | If the edge is part of the minimum spanning tree, calculated using Prim's algorithm (Prim, 1957). | $\Theta(n^2)$ |
| NN solution | If the edge is part of solution constructed by the nearest neighbor heuristic. | $\Theta(n)$ |
| NI solution | If the edge is part of solution constructed by the nearest insertion heuristic. | $\Theta(n^3)$ |

Table 4: Summary of additional input features evaluated. While more features can help produce better predictions, and thus better guidance for the GLS algorithm, they come at the cost of additional computation time.

We semi-randomly sample 100 input feature sets using Latin hypercube sampling (McKay et al., 1979). We train a model using each of these feature sets for 35 epochs without early stopping and record the final validation loss. Using these results, we fit a Gaussian process to emulate the mapping between the feature set $F = \{f_0, f_1, \ldots, f_{10}\}$, and the validation loss $l$ achieved by our model after training, where $f_n$ indicates whether or not feature $n$ is used. We then compute Monte-Carlo estimates of the main and total effects of each input feature on the model's validation loss (Saltelli, 2002). Our implementation uses Emukit (Paleyes et al., 2019). Figure 6 depicts the estimated total effect for each input feature. Edge weight is the most important feature, followed by neighbor rank, depot weight, edge width, and node width. The remaining features have little to no importance.

We train a model using these top five features on 20-node problems and compare its performance to the equivalent model using edge weight as the only feature when solving the 20, 50, and 100-node problem sets. The performance of both models at the computation time limit is shown in Table 5. While the model using additional input features produces slightly more accurate regret predictions, any benefit is cancelled out by the additional time required to compute these features. Note that the results are slightly different from those reported in Figure 5 due to different training hyperparameters.

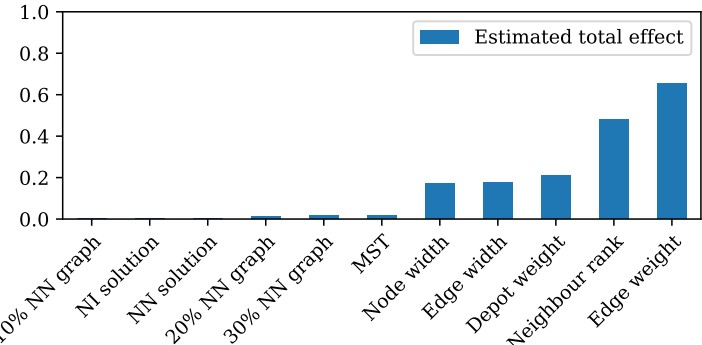

Figure 6: The estimated total effect of edge weight and ten additional features on model validation loss after training based on 100 randomly-sampled feature sets, using Gaussian process-based global sensitivity analysis. A small effect implies that the feature is not important when predicting regret, and vice-versa.

| Problem | Method | Optimality gap (%) | Optimal solutions (%) |
|---|---|---|---|
| TSP20 | Local search only | 1.824±3.252 | 54.7000 |
| | Edge weight only | **0.000±0.000** | **100.0000** |
| | Top 5 features | 0.000±0.003 | 99.9000 |
| TSP50 | Local search only | 3.357±2.602 | 6.4000 |
| | Edge weight only | 0.082±0.254 | 82.1000 |
| | Top 5 features | **0.080±0.295** | **83.2000** |
| TSP100 | Local search only | 4.169±2.046 | 0.3000 |
| | Edge weight only | **2.183±1.383** | **2.2000** |
| | Top 5 features | 2.228±1.421 | **2.2000** |

Table 5: Ablation analysis of our method using local search alone (no guidance), a model using edge weight alone, and model using additional features. Both models are trained on 20-node problems and evaluated on the 20, 50, and 100-node problem sets. The mean optimality gap and standard deviation, as well as the percentage of optimally solved problems are reported after 10 seconds of computation time per instance. While the model using additional input features produces slightly more accurate regret predictions, any benefit is cancelled out by the additional time required to compute these features.

