# OpenReview forum: "Graph Neural Network Guided Local Search for the Traveling Salesperson Problem"
_ICLR.cc/2022/Conference — ICLR 2022 Poster_

### Official Review · Reviewer_u4N8 · 2021-10-30

**Correctness:** 2
**Technical Novelty And Significance:** 2
**Empirical Novelty And Significance:** 2
**Recommendation:** 3
**Confidence:** 5

**Main Review:**

Pros:

1. The paper is well-written and easy to follow.

2. Hybridizing machine learning with local search is an interesting approach.

3. The experimental results show that the proposed algorithm outperforms other recently published learning-based algorithms, which is nice.


Cons:

1. The motivation of this work is not strong enough. The paper is motivated by "solving large TSP instances quickly without sacrificing solution quality remains challenging for current approximate algorithms". Since there exist exact solvers such as Concorde that can solve large-scale TSPs quite efficiently, the authors should further justify why improving approximate algorithms for TSP is interesting.

2. The motivation is not well-supported by the experiments. The paper only presents the results for TSP up to 100 nodes, and no evidence is provided to show that the method can perform well for large-scale TSPs.

3. Only learning-based methods and a weak baseline GLS are compared in this paper. Some effective traditional methods for TSP such as LKH are missing. The exact method Concorde should also be compared since it is generally very fast for solving TSP instances with a few hundred nodes.

4. The proposed method is only evaluated on randomly generated instances. In literature, there are standard benchmark instances such as those collected in the TSP library, which are much harder than randomly generated instances. The proposed method should also be evaluated on standard benchmark instances in order to show its efficacy.


**Summary Of The Paper:**

The paper presents a machine learning approach to improve Guided Local Search (GLS) for finding high-quality solutions for the Travelling Salesman Problem (TSP). Specifically, the paper proposes to use an ML model to predict the global regret for an edge, i.e., the difference in the objective values of a locally optimal solution with that edge being included and a globally optimal solution. Then, the prediction is used to guide the GLS to escape local optimal solutions.


**Summary Of The Review:**

Although the paper shows some improvements over several learning-based methods on small TSP instances, more comparison between the proposed method and effective traditional algorithms on large-scale TSP instances is required in order to support the claim of the paper.

---

> ### Author Response · Authors · 2021-11-23
> **Response to reviewer feedback**
>
> We appreciate the reviewer’s comment that the motivation for improving approximate methods for the TSP is not strong enough. We have added to the introduction and Section 2 (Related Work), paragraph 2 to highlight that while Concorde and LKH-3 are very effective and efficient solvers for the TSP and VRPs, they are not typically used in industry for solving real-time routing problems. Practical problems often impose constraints on top of the basic TSP or CVRP problem definitions and highly specialized solvers are difficult to adapt to new constraints. Thus, there is a need for *general* algorithmic frameworks that can produce high-quality solutions with minimal computation time, which our approach aims to address. Such general frameworks can be easily adapted to various problem definitions with additional constraints while remaining performant.
>
> We have added additional experimental results, demonstrating the performance of our approach and the other learning based approaches when generalizing from the training distribution (consisting of randomly generated TSPs) to TSPLIB instances, which are meant to represent real-world problems. These results are presented in Table 3 (Appendix C). We evaluate our approach and other learning based approaches using models trained on random 100-node TSPs and evaluated on TSPLIB instances with 50 to 200 nodes and 2D Euclidean distances (29 instances). We report mean computation time and optimality gap averaged over 10 runs per instance. Our approach remains on the Pareto front.
>
> We have added additional experiments where both solution quality and computation time are unfixed, shown in Table 1 and Figure 3 (Appendix B1), including results for classical baselines such as Concorde and LKH-3, as well as greedy heuristics (Nearest Neighbor, Farthest Insertion). We have also re-run our experiments for several learning based approaches (Kool et al., Joshi et al.) using their original implementations, which resulted in performance improvements for these approaches. The updated results are shown in Table 1 and 2, as well as Figures 3, 4 (Appendix B), and 5 (Appendix C).
>
> Finally, we have referenced some additional recent approaches for solving the TSP. We have added to Section 2 (Related Work) to highlight the differences between these methods and our work, and to Section 6 (Discussion) in the discussion regarding symmetry.

---

> > ### Comment · Reviewer_u4N8 · 2021-11-29
> > **Response to Authors**
> >
> > Thanks for the response. I don’t think “Concorde and LKH-3 are not typically used in industry for solving real-time routing problems” is a good argument. In fact, the proposed method is even slower than Concorde and LKH-3 (according to Table 1), hence making it less applicable to real-time routing problems. Further, if the goal of this paper is to solve real-time routing problems with new constraints on top of TSP/CVRP, it is important to test the proposed method on those problems instead of TSP. Within the current scope of this paper (TSP), the proposed method is clearly not comparable to baselines such as LKH-3 and Concorde, and the methods are not evaluated on larger problem instances (which was the main motivation of the paper). Hence, I still think this paper is not ready for ICLR.

---

> > > ### Author Response · Authors · 2021-11-29
> > > **Response to reviewer feedback**
> > >
> > > Just as in [1 - 5], we are making progress towards building better (meta)heuristics for graph problems such as the TSP but are not yet faster than Concorde or LKH-3.
> > >
> > > We compare to many recent approaches that use ML to build these solvers, and we beat all of them by a large margin. We did not compare to [4] and [5] but could include them in Table 1 with a note that we cannot know if they lie on the Pareto front with the results provided by the authors (see our discussion [here](https://openreview.net/forum?id=ar92oEosBIg&noteId=QyPSjKlzo_k)). While at this point we cannot be faster than Concorde or LKH-3 on small problems, we hope that this paper serves as a stepping stone towards that goal. As a result, we feel that a rating of 3 is very much unjustified.
> > >
> > > [1] Kool, W., van Hoof, H., & Welling, M. (2018). Attention, Learn to Solve Routing Problems! 7th International Conference on Learning Representations, ICLR 2019. https://arxiv.org/abs/1803.08475v3.
> > >
> > > [2] Joshi, C. K., Laurent, T., & Bresson, X. (2019). An Efficient Graph Convolutional Network Technique for the Travelling Salesman Problem. https://arxiv.org/abs/1906.01227.
> > >
> > > [3] O. da Costa, R., Rhuggenaath, J., Zhang, Y., & Akcay, A. (2020). Learning 2-opt Heuristics for the Traveling Salesman Problem. https://arxiv.org/abs/2004.01608.
> > >
> > > [4] Kool, W., Van Hoof, H., Gromicho, J., & Welling, M. (2021). Deep Policy Dynamic Programming for Vehicle Routing Problems. https://arxiv.org/abs/2102.11756.
> > >
> > > [5] Fu, Z.-H., Qiu, K.-B., & Zha, H. (2021). Generalize a Small Pre-trained Model to Arbitrarily Large TSP Instances. https://arxiv.org/abs/2012.10658v2.

---

> > > > ### Comment · Reviewer_2Y2p · 2021-11-29
> > > > **Misleading claim**
> > > >
> > > > > "We compare to all prior approaches that use ML to build these solvers, and we beat all of them by a large margin."
> > > >
> > > > Have you compared to [4] and [5] directly? If not, it is misleading to claim that your proposed method "beats all of them by a large margin". (In fact, [5] is the current state of the art in terms of optimiality gap.)

---

> > > > > ### Author Response · Authors · 2021-11-29
> > > > > **Response to reviewer feedback**
> > > > >
> > > > > We included [4] and [5] in the list of other papers which do not compare directly to Concorde or LKH-3.
> > > > >
> > > > > We have not compared to [4] and [5] and we cannot compare them directly with results reported by these authors, due to the difference in evaluation methods. We could include them in Table 1 with a note that we cannot know if they lie on the Pareto front (as seen in [our response above](https://openreview.net/forum?id=ar92oEosBIg&noteId=QyPSjKlzo_k)).

---

> > > > > ### Author Response · Authors · 2021-11-29
> > > > > **Response to reviewer feedback**
> > > > >
> > > > > We have edited our original comment to be clearer.

---

> > > > ### Comment · Reviewer_u4N8 · 2021-11-29
> > > > **Response**
> > > >
> > > > It is not supervising that the proposed method outperforms [1,2] by a large margin, because the proposed method is GNN+LS while [1,2] are GNN only. The proposed method is more like [3] (ML+2-opt), and as expected the performances of the proposed method and [3] are similar. In terms of methodology, enhancing local search/heuristic by machine learning is not a new idea (there are several papers on this already, including [3]). In terms of performance, the proposed GNN+LS is struggling to compete with strong baselines, e.g., Concorde and LKH (which is also a "local search"). Note that the exact solver Concorde can solve TSP instances with thousands of nodes, while the largest instance tested in this paper is about 200. If the goal of this paper is to solve real-time routing problems with new constraints (as stated by the authors), the proposed method needs to be adapted to and evaluated on those problems. Hope this better clarifies my score.

---

> > > > > ### Author Response · Authors · 2021-11-29
> > > > > **Response to reviewer feedback**
> > > > >
> > > > > Our approach is most similar to [2], not [3], as it produces predictions on edges and uses another algorithm to build solutions. Our approach uses GLS (a metaheuristic) and [2] uses beamsearch. [3] repeatedly queries an ML model to make decisions. Following [6], it falls into the first or third class of machine learning methods for combinatorial optimization, where our approach falls into the second class.
> > > > >
> > > > > Many prior works produce predictions on edges and construct tours using heuristics [2, 4, 5]. To our knowledge, we are the first to use predictions to inform a *metaheuristic*.
> > > > >
> > > > > Our approach totally dominates [3] (i.e. it produces better solutions and faster) when solving randomly generated TSPs and TSPLIB instances (see Table 2, and [here](https://openreview.net/forum?id=ar92oEosBIg&noteId=G_b3PDbhKmx) for average performance on TSPLIB instances).
> > > > >
> > > > > Regarding Concorde and LKH-3, our claims and contributions are in line with the community's current values to include results from these methods, but to separate the evaluation of ML approaches (as seen in [1 - 5]). From *Guidelines for the Computational Testing of Machine Learning approaches to Vehicle Routing Problems* [7]:
> > > > > > Despite the purpose of ML-based approaches not being that of outperforming highly specialized solvers, but rather that of proposing versatile tools not requiring high-levels of manual engineering, the comparison should still occur against the best performing algorithms to better comprehend the tradeoff between data-driven and ad-hoc algorithms.
> > > > >
> > > > > [6] Bengio, Y., Lodi, A., & Prouvost, A. (2020). Machine learning for combinatorial optimization: A methodological tour d’horizon. In European Journal of Operational Research. https://doi.org/10.1016/j.ejor.2020.07.063
> > > > >
> > > > > [7] Accorsi, L., Lodi, A., & Vigo, D. (2021). Guidelines for the Computational Testing of Machine Learning approaches to Vehicle Routing Problems. https://arxiv.org/pdf/2109.13983.pdf

---

> ### Author Response · Authors · 2021-11-26
> **Response to reviewer feedback**
>
> We thank the reviewer again for the detailed review. We believe we have addressed all the concerns that were raised (as described in our detailed rebuttal). In particular, we contextualized our work with respect to classical approaches, have added classical baselines, and evaluated our method on the TSPLIB benchmark problem set. Our approach dominates other ML-based approaches (it is faster and produces better solutions) when solving the uniform random problem sets and when solving the TSPLIB instances (see Tables 1 and 3). Please let us know if you need any further clarification.
>
> Average performance across 29 TSPLIB instances:
>
> | Metric                | Kool et al.   | Joshi et al.  | da Costa et al. | Ours         |
> |-----------------------|---------------|---------------|-----------------|--------------|
> | Mean computation time | **0.532±0.300**   | 7.000±2.415   | 31.766±4.543    | **10.420±0.630** |
> | Mean optimality gap   | **16.767±67.953** | 40.025±15.702 | 1.725±3.767     | **1.529±1.328**  |
> |                       |               |               |                 |              |
>
> Kool’s and our methods are Pareto optimal, and the others are totally dominated.

---

### Official Review · Reviewer_v3Sn · 2021-10-31

**Correctness:** 3
**Technical Novelty And Significance:** 3
**Empirical Novelty And Significance:** 3
**Recommendation:** 8
**Confidence:** 3

**Main Review:**

In recent years, methods for the TSP have paid little attention to the fairness of the evaluation criterion, and this paper achieved a good performance under the improved evaluation method. But I suppose the experiment should be more comprehensive.

Pros:

1. The author used edge length as the feature when designing the network, and analyzed the pros and cons of each network in detail in the Discuss section. Such detailed analysis is extremely necessary.

2. They chose a better way to evaluate, which is valuable, because the evaluation methods in recent years lack in this respect. The previous evaluation method is easy to find a loophole.

Cons:

1.	The ablation study is not detailed enough. The method proposed by the author is based on GLS, but uses LS as the baseline, and the experimental do not show which part of the method contributes more.

2.	The number of cities considered by recent methods is getting larger and larger, far more than 100. So how does the method proposed by the author perform on instances with a larger number of cities, such as 500. Will Global regret become unpredictable?



**Summary Of The Paper:**

This paper proposed an approach for solving TSP, using GLS guided by global regret, and designed a neural network to predict the regret. The authors evaluates the performance of each method by calculating gap between the prediction and the global optimum in a fixed computation time.

**Summary Of The Review:**

The contribution of the paper is significant and somewhat new with insights into the TSP problem by GNN.

---

> ### Author Response · Authors · 2021-11-23
> **Response to reviewer feedback**
>
> We thank the reviewer for their comments. To address these we have:
> 1. Added the performance of local search alone in Table 4 (Appendix D) as part of the ablation study. The reader can directly compare the performance with local search alone, local search guided by our regret prediction model using edge weight only, and local search guided by our regret prediction model using the top five features.
> 2. Evaluated the performance of our approach and the other learning based approaches when generalizing from the training distribution (consisting of randomly generated TSPs) to TSPLIB instances, which are meant to represent real-world problems. These results are presented in Table 3 (Appendix C). We evaluate models trained on random 100-node instances on TSPLIB instances with 50 to 200 nodes and 2D Euclidean distances (29 instances). We report mean computation time and optimality gap averaged over 10 runs per instance.
>
> We have adjusted the evaluation method to align better with recent computational guidelines for learning based approaches for the TSP. Specifically, we have added additional experiments where both solution quality and computation time are unfixed, shown in Table 1 and Figure 3 (Appendix B1). These allow us to evaluate other approaches in a way that is closer to their original implementation. We have also re-run our experiments for several approaches (Kool et al., Joshi et al.) using implementations that are closer to their original implementations, which resulted in performance improvements for these approaches. We have also added classical baselines to the results in Table 1, including well known and highly effective traditional algorithms such as Concorde and LKH-3, as well as greedy heuristics (Nearest Neighbor, Farthest Insertion). The updated results are shown in Table 1 and 2, as well as Figures 3, 4 (Appendix B), and 5 (Appendix C).
>
> We have also added to the introduction and Section 2 (Related Work) to strengthen our motivation for this work, which is not to outperform highly specialized TSP and VRP solvers like Concorde and LKH-3, but to develop general algorithmic frameworks that can produce high-quality solutions with minimal computation time. Finally, we have referenced some additional recent approaches for solving the TSP. We have added to Section 2 (Related Work) to highlight the differences between these methods and our work, and to Section 6 (Discussion) in the discussion regarding symmetry.

---

### Official Review · Reviewer_2Y2p · 2021-11-02

**Correctness:** 3
**Technical Novelty And Significance:** 3
**Empirical Novelty And Significance:** 2
**Recommendation:** 6
**Confidence:** 5

**Main Review:**

# 1. Strengths

1.1. To the best of my knowledge, this is the first work on **combining deep learning with a metaheuristic** (GLS).  The proposed idea of using a GNN to learn the global regret to guide the LS + GLS search procedure  is novel and interesting.

1.2. The approach of **operating on the line graph of routing problems** like TSP is somewhat novel (but may need to be better contextualized + ablated, see suggestions below). This is potentially more principled than the prevalent approach of encoding node coordinates, which does not account for the underlying symmetries of routing problems. The discussion on this aspect in Section 6 is thought-provoking.

1.3. The paper proposes a new evaluation setup, which aims to **evaluate trade-offs between solution quality and time**. This effort to move away from somewhat simplistic evaluations on fixed TSP sizes without a time limit is appreciated, as it provides more depth to the results. A caveat is that I do have some issues with the proposed setup that need to be addressed (see below).

---

# 2. Weaknesses & Things to be addressed

## 2.1. On evaluation setup:
- 2.1.1. I agree with the authors that the current style of evaluating deep learning approaches for TSP/routing has several pitfalls and does not give the full picture. Thus, I do appreciate their introduction of a fixed computation time budget per test instance and %-age of optimally solved instances into the picture. The evaluation of deep learning approaches for routing has recently been questioned by the community, too [8, 9].
- 2.1.2. **How to contextualize empirical improvements without standard setup?**: The current set of experiments and choice of baselines did not convince me sufficiently about the empirical improvements being brought about by the novel methodology, as recent papers have significantly improved over Kool-etal, Joshi-etal, e.g. [5, 6, 7]. In addition to the proposed evaluation setup, I would still have liked to see the paper compare to existing literature in the conventional format of reporting optimality gap, tour length and total inference time for 10000 instances of TSP20, 50 and 100. Alternatively, the paper could consider showing results for generalization to real-world TSPLib instances, for which many recent papers are providing tour lengths per instance.
- 2.1.3. **Why fix batch size = 1?**: One of the major advantages of deep learning approaches for combinatorial problems is the ability to perform batched inference. Thus, it was difficult for me to understand why the paper chose to set batch size = 1 during the evaluation, especially as the motivation was real-time inference. Is it possible to implement a batched variant of LS + GSL, or is it a sequential algorithm? Is the search procedure run on the GPU or the CPU?
- 2.1.4. **Why not use beam search in under 10s?**: The extremely poor results for Joshi-etal + Sampling on TSP100 are surprising, as the original paper reported optimality gaps 1.5-2%. I believe it is important to fairly compare to this work as it is most related in terms of methodology (see my comments on related work). Can you also show results for the original formulation (i.e. beam search) of their method? (Based on the results and computational times they have reported, the beam search is critical for getting the model to work and should take <10s per instance, too.)
- 2.1.5. **Why not include classical baselines**: Once again, I believe that the Concorde solver can very easily solve small TSP100 to optimiality within 10s per instance (in fact, I am sure it would be under 1s). Similarly, the LKH-3 heuristic is also not taking longer than 10s per instance. I think it is critical to compare to these classical and well known baselines.

## 2.2. On missing latest related work and comparisons:
- 2.2.1. **Missing one-shot deep learning approaches**: Within the ‘ML alone provides a solution/provides information to an OR algorithm’ sub-section, there are works with sequentially built solutions [2, 3] and those that output the solution in one-shot [4]. I believe this paper is in the second camp: similar to Joshi-etal [4], the neural network outputs predictions over edges, which are then used to build a solution via a classical search algorithm. Thus, it is critical to also contextualize how the specific search algorithm used in this work (LS + GLS) compares to recent/concurrent advances on top of Joshi-etal’s model: e.g. leveraging dynamic programming [5] or MCTS [6] to replace the original beam search.
- 2.2.2. **Missing sequential approach POMO**: An important reference that is missing is POMO [7], a recent NeurIPS paper which can be considered an update to Kool-etal [3] from the lense of symmetries in solutions for routing problems. Including POMO in the related work + potentially in the experimental comparisons will further strengthen the paper. POMO should also be discussed in the Discussions section at the end of the paper regarding its use of symmetries, and how it may be different from the proposed approach.

## 2.3. Missing generalization to other problems
An important limitation which, to the credit of the authors, is openly pointed to is that the work only shows results for TSP. The paper would be significantly strengthened by demonstrating at least some degree of generality of the proposed approach, e.g. CVPR could be the first candidate. It is not convincing to simply say that the method can be applied. (Although I do understand that this may be challenging to show convincingly within the rebuttal period.)

---

# 3. Other minor comments and suggestions

## 3.1. On message-passing over line graphs vs. message-passing over nodes and edge features:
- 3.1.1. **Wrong claim?**: It may be worth expanding on how different it is to do GNN/message passing over the transformed line graph (as in this work) vs. message passing over the original graph while also including and updating features over edges (as proposed in [4] and adapted by [5, 6]). This has also been done in other application domains, e.g. the famous ChemProp model. The claim that *‘most approaches to edge-property prediction predict the properties of an edge as a function of the surrounding node states’* may not be true, as [4, 5, 6] are explicitly using the edge features to make predictions.
- 3.1.2. **Missing ablation**: Currently, the empirical evaluation only shows results for the entire line-graph GNN + GLS technique. In order to ablate the impact of the line graph transformation and the significance of the proposed GNN architecture, it may be useful to show experiments on replacing the proposed line-graph GNN with: (1) GNN on original graph w/ and w/out edge feature updates; (2) simple MLP applied to each graph edge’s original feature.

## 3.2. On motivation for this work:
- 3.2.1. In the abstract, the major motivations for the proposed techniques is to approximately solve large-scale TSP instances **and** to provide solutions in real-time without sacrificing solution quality.
- 3.2.2. **Generalization to real-world sizes**: More experiments on scalability of training or zero-shot genreralization to TSPs bigger than 100 nodes or on real-world instances from TSPLib would make the claims more convincing.
- 3.2.3. **Real-time and fast routing solvers**: The introduction briefly mentions that this remains a challenge for current algorithms which do not regard computation time. To make this claim stronger, it may be useful to point out how real-time routing problems are typically being solved in industry and why there exists a research gap that this paper is attempting to fill. (e.g. KGLS [1] exists but is it requiring a lot of handcrafting?)

## 3.3. On scalability of dataset generation:
It was noted in [8] that generating supervised datasets is a limitation in scaling up, as building these datasets itself can be computationally challenging for real-world problem scales beyond 100s of nodes. Isn’t developing the training datasets for your approach further more cumbersome than previous supervised approaches, e.g. your approach requires running Concorde as well as multiple runs of LKH3. Is this approach scalable beyond small TSP100 instances (especially considering that one may need more training data to train on harder problems)?

## 3.4. On style of reporting empirical results:
I personally found the style of reporting performance improvements in the abstract and contributions bullets in the introduction to be rather confusing. I would suggest that the paper sticks to either using percentage improvements (x% improvement over y), times improvements (x times improvement over y), but not using both. It could even be simpler and more informative to rephrase something like ‘we reduce the average optimality gap on TSPn from x_1 to x_2, a y times/%-age improvement’. This will help experts contextualize the results immediately.

---

# References

[1] Florian Arnold and Kenneth Sorensen. Knowledge-guided local search for the vehicle routing problem. Computers & Operations Research, 105:32–46, 2019.

[2] Michel Deudon, Pierre Cournut, Alexandre Lacoste, Yossiri Adulyasak, and Louis-Martin Rousseau. Learning heuristics for the TSP by policy gradient. In Integration of Constraint Programming, Artificial Intelligence, and Operations Research, pp. 170–181, 2018.

[3] Wouter Kool, Herke Van Hoof, and Max Welling. Attention, learn to solve routing problems! arXiv preprint arXiv:1803.08475, 2018.

[4] Chaitanya K Joshi, Thomas Laurent, and Xavier Bresson. An efficient graph convolutional network technique for the travelling salesman problem. arXiv preprint arXiv:1906.01227, 2019.

[5] Wouter Kool, Herke van Hoof, Joaquim Gromicho, Max Welling. Deep Policy Dynamic Programming for Vehicle Routing Problems. arXiv preprint, 2021.

[6] Zhang-Hua Fu, Kai-Bin Qiu, Hongyuan Zha. Generalize a small pretrained model to arbitrarily large tsp instances. AAAI 2021.

[7] Yeong-Dae Kwon, Jinho Choo, Byoungjip Kim, Iljoo Yoon, Youngjune Gwon, Seungjai Min. POMO: Policy Optimization with Multiple Optima for Reinforcement Learning. NeurIPS 2020.

[8] Chaitanya K. Joshi, Quentin Cappart, Louis-Martin Rousseau, Thomas Laurent. Learning TSP Requires Rethinking Generalization. CP 2021.

[9] Luca Accorsi, Andrea Lodi, Daniele Vigo. Guidelines for the Computational Testing of Machine Learning approaches to Vehicle Routing Problems. arXiv preprint, 2021.


**Summary Of The Paper:**

This paper proposes a novel deep learning + guided local search heuristic technique for approximately solving the Travelling Salesperson Problem (TSP). This research direction is **well motivated**, as inexpensive but approximate heuristics to graph combinatorial optimization problems such as TSP are promising for enabling real-time applications in vehicle routing and operations research.

From a methodological perspective, the paper’s major contribution is to develop and train a **Graph Neural Network (GNN) on line graphs** for approximating the ‘regret’ of including graph edges in TSP solutions. The use of regret per edge is novel; regret is a metric used in the metaheuristic **Guided Local Search (GLS)**. After the GNN is trained, the proposed technique starts from a greedy solution and uses a TSP-specific Local Search interleaved with the **‘learnt’ GLS metaheuristic** to arrive at the final solution.

From an empirical point of view, this work proposes a **new evaluation setup** to measure solution quality of TSP heuristics within a fixed computational time. Their GNN + GLS approach outperforms popular learnt baselines and handcrafted GLS in this setup for TSPs up to 100 nodes.

**Summary Of The Review:**

I recommend that this paper is currently **marginally below the acceptance threshold**. Overall,  I found the methodological ideas (learning-guided metaheuristics + GNNs on line graphs to respect the symmetry of combinatorial problems) to be both novel and interesting. However, I have major issues with the **lack of extensive experimental evaluation**. In particular, the experiments are insufficient to convince the reader that the proposed methodology leads to significant empirical improvements over recent works, or can generalize to real-time routing problems beyond the TSP. It is also difficult to contextualize this paper’s results as the evaluation setup is not conventional and comparissons to classical baselines like Concorde and LKH3 are absent. While the new setup may be a step in a positive direction, additional results **comparing to the latest published work** [6,7] in a conventional format or generalization to real-world and **larger scale TSPLib instances** would be needed for me to be convinced to improve my score. If the major motivation for the work is to tackle large-scale routing problems in real-time, demonstrating that the proposed techniques work on TSPs larger than just 100 cities and on non-random distributions is also important.

---

# Update after rebuttal

I appreciate the updates to the evaluation setup and the reported results. I also appreciate the inclusion of TSPLib experiments and discussions on recent literature [5,6,7,9]. Most of my concerns are addressed, and I am happy to **raise my score to an accept** after the authors provided clarifications to the newly introduced changes (discussed below in the comments).

I believe this work is **bringing new methodological ideas** to the literature on learning-driven combinatorial optimization and will be of interest to the community. However, I am still lukewarm because the motivation of the work is to tackle real-time routing problems with non-standard constraints, but the **empirical results are only shown for TSP**. In the paper's present state, it is not clear to me whether this approach will generalize to other routing problems, especially those problems with more challenging constraints for which generating labelled training datasets is an issue.

---

> ### Author Response · Authors · 2021-11-23
> **Response to reviewer feedback**
>
> We thank the reviewer for their comments and suggestions.
>
> ## On evaluation setup
> We appreciate the reviewer’s reference to [9] and agree with the guidelines given by these authors. We have modified our experimental setup to be as in line with these guidelines as possible.
>
> In addition to our fixed computation time experiments, we have added another set of experiments more aligned with the traditional evaluation method: we leave solution quality and computation time unfixed and compare approaches as-described by the authors. We report mean and standard deviation for optimality gap and computation time per instance, measured across the entire problem set. These results are shown in Table 1, and visualized in Figure 3. This evaluation method (computation time per instance versus optimality gap) is recommended by [9] and is common in OR literature. We believe that measuring computation time per instance is more meaningful than the time to solve an entire problem set in practical scenarios. We have also added classical baselines (Concorde, LKH-3, as well as greedy construction heuristics: Nearest Neighbor, and Farthest Insertion) to our evaluation in Table 1 and Figure 3.
>
> We agree that leveraging parallelism in the GPU is an important advantage of deep learning-based approaches for combinatorial optimization. We have modified the experimental setup to solve one problem instance at a time, but leverage parallelism in the GPU to sample multiple solutions to the same instance at once, where implementations supported this. This resulted in better performance for Kool et al., especially in generalization, (see Figure 4). A detailed description of the configurations used for each approach is included in Appendix A. We argue that parallelism in the GPU should not be used to solve multiple problems in a single batch: in practical implementations, a sensible, and more efficient design choice would be to run multiple processes on the GPU rather than batching problems and solving them on a single thread. Therefore, it is not realistic to solve multiple problems in a single batch. Furthermore, solving a single problem at a time allows us to measure computation time per instance, which is recommended by [9] and is more aligned with existing OR literature.
>
> We have changed all Joshi et al. results to use beamsearch rather than the sampling approach. This improved performance for their method (see Tables 1 and 2, and Figures 3 and 4). We found a programming error in the original beamsearch implementation that produced invalid tours (the depot node is visited more than once, and one node is not visited). We ignored these invalid tours, so our results do not correspond to the results originally reported by the authors. Our fix is shown in Appendix A.
>
> ## On missing related work and comparisons
> We have added [5, 6] to the related work and highlighted the differences between these methods and our work in section 2. These approaches construct tours using heuristics, whereas we are the first to use a metaheuristic. We have added [7] to the related work, and referenced it in the discussion regarding symmetry. We prioritized adding classical baselines and using the evaluation methods recommended in [9] for our approach, the new baselines, and the learning methods (see Table 1 and Figure 3), and evaluating generalization for our approach and other learning methods to TSPLIB (see Table 3, Appendix C) over adding a comparison of these recent works.
>
> ## Missing generalization to other problems
> Thank you for your comment, we will certainly pursue this as an extension to this work.
>
> ## Other minor comments and suggestions
> *  Thanks for the comments at 3.1. We have added related literature in Related Work and updated our statement. Our approach differs from prior work [4, 5, 6] in that our model does not require node features. Without transforming the original graph, place-holder features would be needed at each node, which we believe is a less principled approach.
> We also demonstrated an ablation study of input features in Appendix D.  Our intuition for MLP is that it will lose generalization capability as the size of the trainable parameter depend on the input, but the permutation invariant of GNN makes it able to scale and generalize, as shown in Table 3. We will leave the cross-comparison with MLP into future work for the interest of time.
> * We added results on generalizing from random TSPs to TSPLIB instances. We trained on 100-node random TSPs and evaluated all 50 to 200 node TSPLIB instances (29 in total) The results are shown in Table 3.
> We added language in the introduction and section 2, paragraph two to emphasize the gap that our approach and KGLS attempt to address.
> * We agree that building training sets for larger problem instances using our approach is cumbersome. We added language discussing this to the last paragraph of section 6.
> * We have modified our results reporting to only use “Y times improvement”.

---

> > ### Comment · Reviewer_2Y2p · 2021-11-25
> > **Reviewer's response to rebuttal (Part 1)**
> >
> > I appreciate the updates to the evaluation setup and the reported results. I also appreciate the inclusion of TSPLib experiments and discussions on recent literature [5,6,7,9].
> >
> > Most of my concerns are addressed, and I am happy to raise my score to an accept upon some clarifications to the newly introduced changes (see below).
> >
> > I believe this work is bringing new methodological ideas to the literature on learning-driven combinatorial optimization and will be of interest to the community. However, I am still lukewarm because the motivation of the work is to tackle challenging routing problems with non-standard constraints, but the empirical results are only shown for TSP. It is not clear to me whether this approach will generalize to other routing problems at its present stage, especially those problems with more challenging constraints for which generating labelled training datasets is an issue.
> >
> > # Things that need clarifications
> >
> > ### Motivation to develop general frameworks for routing problems
> >
> > > "We agree that building training sets for larger problem instances using our approach is cumbersome. "
> >
> > How do you reconcile the above statement (i.e. the need to generate high-quality supervised training data) with your comment on the paper's motivation to the other reviewers as well as your comment summarizing your rebuttal:
> >
> > > "We acknowledge that Concorde and LKH-3 can easily solve 100-node TSPs. However, we argue that Concorde and other highly-specialized solvers are not used to solve real-time routing problems in industry because practical problems often impose constraints on top of the basic TSP or CVRP problem definitions. Highly specialized solvers are difficult to adapt to new constraints. Thus, there is a need for general algorithmic frameworks that can produce high-quality solutions with minimal computation time, which our approach aims to address. Such general frameworks can be easily adapted to various problem definitions with additional constraints while remaining performant."
> >
> > It is not clear to me whether the proposed approach actually addresses the issues you have highlighted, namely 'highly specialized solvers are difficult to adapt to new constraints' while a 'general framework can be easily adapted to various problem definitions with additional constraints'.
> >
> > Based on my understanding, if the proposed approach were to be used to tackle a challenging routing problem beyond TSP/CVPR with non-standard constraints, one would firstly need to generate a supervised training dataset for which one needs exact solvers in the first place (which may or may not be available). Then, one would need to design problem-specific local search operators for LS+GLS. Thus, applying the proposed approach to new and challenging routing problems seems to require significant effort.
> >
> > The workflow seems to be: build an exact solver -> generate labelled training data -> define good local search operators -> train proposed approach.
> >
> > Could you comment on the 'generality' of such a framework?
> >
> > ---
> >
> > ### Not reporting performance for [6, 7]
> >
> > > "We prioritized adding classical baselines and using the evaluation methods recommended in [9] for our approach, the new baselines, and the learning methods (see Table 1 and Figure 3), and evaluating generalization for our approach and other learning methods to TSPLIB (see Table 3, Appendix C) over adding a comparison of these recent works."
> >
> > **In Table 1, is there a reason why your results cannot be directly compared to [6, 7], at least in terms of optimality gap**
> >
> > I understand that these papers have reported computational time in a different way (and agree that reporting per instance time may be more informative), but I believe that the optimality gaps on random TSP20, 50, 100 instances may still be directly comparable. Both [6, 7] have reported significantly better optimality gaps on TSP100 with the same evaluation procedure: 0.03% and 0.14%, respectively, compared to this paper's 0.68%. Thus, I would like to understand why these results are omitted from Table 1.
> >
> > (While I understand that it may not be possible to re-run each and every model on this paper's specific randomly generated 10K TSP test instances, it may still be helpful to the reader to report performance of SotA models with an asterixis (*) denoting that the results are directly imported from the respective papers.)
> >
> > ---
> >
> > (continued in [Part 2](https://openreview.net/forum?id=ar92oEosBIg&noteId=D0BcAHRPMCS))

---

> > > ### Author Response · Authors · 2021-11-26
> > > **Response to reviewer feedback**
> > >
> > > ## Motivation to develop general frameworks for routing problems
> > >
> > > Our approach is really geared towards real-time routing scenarios, where we want to find good solutions quickly. This remains difficult for current approaches, and it is the gap that our work and KGLS [1] is trying to fill. In “Such general frameworks can be easily adapted to various problem definitions with additional constraints *while remaining performant*.”, we should have stated “*while continuing to produce high-quality solutions under restricted computation time*”. This is what we have added in the introduction: “*There is a need for general, anytime combinatorial optimization algorithms that produce high-quality solutions under restricted computation time*”.
> > >
> > > To produce good solutions quickly, our approach essentially shifts the computation time to training time: the training set can be produced offline, with any solver, and with unrestricted computation time. It is *computationally* cumbersome, but does not require the designer to invest time in tuning the algorithm. Our proposed workflow would be:
> > > 1. Develop a GLS+LS algorithm with the appropriate operators.
> > > 2. Use this algorithm (not guided by regret) to generate a training set: there is no computation time constraint at this stage, so we can allow the algorithm to run as long as we like.
> > > 3. Train the regret prediction model.
> > > 4. In real-time, use the regret prediction model to guide the same GLS algorithm towards good solutions, much faster than was originally possible.
> > >
> > > When generating the training set, we believe that an exact solver is not necessary - we used LKH-3 to calculate the fixed-edge tours. In fact, it is likely that LKH-3 did not always find optimal fixed-edge tours when computing the training set: we used the same configuration to compute the training set as we used to compute the results in Table 1, where LKH-3 did not find optimal tours for all problems. So, it is not essential for the training set to be very high-quality.
> > >
> > > One drawback of our method is the need to define a set of local search operators. However, we believe that our approach of “a general metric that is defined for all problems” and “plug-and-play a set of local search operators” transfers more easily to other problems compared to other ML methods. For example, da Costa’s method will not perform well on problems where 2-opt is not effective. One example of a problem like this is the Pickup Drop-off TSP (PDTSP), essentially a problem where packages must be picked up and delivered on the tour. 2-opt cannot be used because it reverses the intermediate section of the tour, which violates the constraint that pickup must precede drop-off (see [10] section 3.4). On the other hand, construction-based methods (like Kool's) can use an action masking method to take these constraints into account, but they must sample to improve on their solution. We can add this consideration in the discussion section.
> > >
> > > Regarding the computational burden of producing a training set, we find it encouraging that our approach generalizes better than others. We think it would actually be feasible to train on smaller problems and generalize to larger ones using our approach, where this was not really possible with other approaches (with the exception of [6]). For example, our approach achieves better results when generalizing from 50 to 100 node problems than da Costa’s method does when *training* on 100 node TSPs (1.378 vs. 1.534 - see Figures 4 and 5).
> > >
> > > ## Not reporting performance for [6, 7]
> > >
> > > We *could* add the results for [6, 7] to Table 1, but would need to add a note that we cannot know if they lie on the Pareto front because we have no computation time per instance results for them. We argue that one should never compare a single metric (computation time or optimality gap) directly, as there is a fundamental trade-off between them. In Table 1 we stopped our method after 10s of computation time, however the results would continue to improve, perhaps outperforming [6, 7] in terms of optimality gap, if we had let it run longer. **This is the loophole in current evaluation methods for the TSP that we are trying to address with our computation time experiments**, and follows [9] and current OR evaluation methods.
> > >
> > > [10] Exponential-Size Neighborhoods for the Pickup-and-Delivery Traveling Salesman Problem, Pacheco et al. 2021. https://arxiv.org/abs/2107.05189

---

> > > > ### Comment · Reviewer_2Y2p · 2021-11-26
> > > > **Updated the score**
> > > >
> > > > Thank you for clarifying my concerns. I have raised my score in the main review to reflect my updated views on the work.
> > > >
> > > > ---
> > > >
> > > > > "We can add this consideration in the discussion section."
> > > >
> > > > I do believe the paper would benefit from explicitly talking about the advantages as well as drawbacks of the proposed approach compared to other techniques, especially when discussing how one may tackle novel and challenging routing problems.
> > > >
> > > > > "In Table 1 we stopped our method after 10s of computation time, however the results would continue to improve, perhaps outperforming [6, 7] in terms of optimality gap, if we had let it run longer."
> > > >
> > > > It would be better to back any such claims with empirical results.

---

> > ### Comment · Reviewer_2Y2p · 2021-11-25
> > **Reviewer's response to rebuttal (Part 2)**
> >
> > (continued from [Part 1](https://openreview.net/forum?id=ar92oEosBIg&noteId=CBvPVupsGN0))
> >
> > ### TSPLib Experiments
> >
> > > "We added results on generalizing from random TSPs to TSPLIB instances. We trained on 100-node random TSPs and evaluated all 50 to 200 node TSPLIB instances (29 in total) The results are shown in Table 3."
> >
> > I especially appreciate the inclusion of these results. However, I have some quick concerns.
> >
> > (1) Is it possible to provide annotations to the table, e.g. bold the best technique for each row, provide the overall average optimality gap across each column? This would help get a 'big picture' view of these real-world results.
> >
> > (2) In Table 3, I observed that the proposed technique's results are comparable or worse than O. da Costa et al. It is especially evident that da Costa's 'learning to do local search' technique outperforms the proposed technique on the largest instances in consideration, e.g. the last 5 instances of size 159, 195, 198, 200, 200. **How do you interpret the results in Table 3?** Is there a reason why da Costa's technique shows stronger performance on larger real-world instances?

---

> > > ### Author Response · Authors · 2021-11-26
> > > **Response to reviewer feedback**
> > >
> > > ## 1)
> > > In experiments with unfixed computation time and solution quality (Table 1, Table 3) we prefer to bold Pareto optimal values (i.e. those that are faster but produce worse solutions, or are slower but produce better solutions), as seen in [9], Figure 1. We agree that showing an average computation time and optimality gap over all instances would allow the reader to compare different methods at a glance - as seen in [9], Table 1. We will also add a description of the experiment setup in the caption of our table.
> > >
> > > These are the average results:
> > >
> > > | Metric                | Kool et al.   | Joshi et al.  | da Costa et al. | Ours         |
> > > |-----------------------|---------------|---------------|-----------------|--------------|
> > > | Mean computation time | **0.532±0.300**   | 7.000±2.415   | 31.766±4.543    | **10.420±0.630** |
> > > | Mean optimality gap   | **16.767±67.953** | 40.025±15.702 | 1.725±3.767     | **1.529±1.328**  |
> > > |                       |               |               |                 |              |
> > >
> > > Kool’s and our methods are Pareto optimal and the others are totally dominated.
> > >
> > > ## 2)
> > > Our results should not be interpreted as worse. Note that da Costa’s technique takes 30s of computation time per instance while ours takes 10s. In the smaller instances our method totally dominates da Costa's: it is both faster and finds better solutions. For the larger instances, da Costa’s method finds better solutions but is slower, so our method does not totally dominate it. For our own method, we can explain the poorer performance on the larger methods by a poorly implemented local search procedure (implemented in Python). As the instances become larger, the local search operations take longer, thus our regret prediction model has fewer opportunities to provide guidance before the stopping condition is reached.

---

> > > > ### Comment · Reviewer_2Y2p · 2021-11-26
> > > > **Updated the score**
> > > >
> > > > Thank you for clarifying my concerns. I have raised my score in the main review to reflect my updated views on the work.

---

### Official Review · Reviewer_un7S · 2021-11-02

**Correctness:** 2
**Technical Novelty And Significance:** 3
**Empirical Novelty And Significance:** 3
**Recommendation:** 3
**Confidence:** 4

**Main Review:**

I like about the submission that it goes beyond simply trying to learn optimum solutions using GNNs. Instead, it learns optimum TSP solutions, uses that to approximate the consequences of adding an edge to the solution, thus guiding a local search procedure. In that regard, it could be regarded as a sophisticated local search rather than an ML approach.

The main concern I have with the submission is that it states that combinatorial approaches, e.g. from the operations research community, are focused on computing optimum solutions and have little regard for computation time. This is, in my view, not true: Any classical branch-and-bound approach may be run in heuristic mode where it is stopped after a couple of seconds. While this will clearly not guarantee an optimum solution, it will quickly provide a good solutions, since B&B approaches tend to spend most of their running time proving optimality and typically find a good solutions at the very beginning of their execution. As opposed to what the submission states in the introduction, these algorithms will provide a quality guarantee (something that this approach does not), just not a global one. Setting such a timelimit provides precisely the tradeoff between computation time and quality (guarantee) that this submission claims. These algorithms work on instances with millions of nodes, whereas this submission considers instances with 100 nodes.

Even is that is not desired: The Euclidean TSP may be approximated in polynomial time with an error that is arbitrarily close to 1 (Arora, J. of the ACM, 1998). Along more practical lines, Christofides' heuristic quickly yields a 3/2-approximate solution.

These algorithms could be included in the experiments as a reference. The OR community uses the TSPLib benchmark set to evaluate TSP algorithms. I am not convinced that the randomly generated instances in this submission are difficult and would like to see solution time/quality of a simple greedy nearest neighbor heuristic.

**Summary Of The Paper:**

The submission is concerned with developing a heuristic for the traveling salesperson problem (TSP) using graph neural networks an dguided local search. The idea of the approach is to learn the regret incurred by including an edge into the solution and to use this to guide a local search out of local optima. As opposed to previous work, the regret is computed against the global optimum (which is approximated by learning). The submission is limited to the Euclidean TSP variant where we operate on a complete graph whose edge weights are given by Euclidean distances.

**Summary Of The Review:**

In my view, the submission fails to compare against the large body of existing literature for the TSP problem and ignores standard benchmark sets. I do not share the submissions view that the Concorde solver and various heuristics are geared towards finding optimum solutions only. I would also like to point out that there are algorithms that do provide quality guarantees. That being said, I am not sure what ML brings to the table here.

---

> ### Author Response · Authors · 2021-11-23
> **Response to reviewer feedback**
>
> We thank the reviewer for their comments and suggestions.
>
> We certainly agree that exact methods can be highly effective for solving the TSP. Concorde can indeed quickly solve 100-node TSPs and provides the additional benefit of an optimality guarantee. However, we argue that Concorde and other highly-specialized solvers are not used to solve real-time routing problems in industry because practical problems often impose constraints on top of the basic TSP or CVRP problem definitions. Highly specialized solvers are difficult to adapt to new constraints. Thus, there is a need for general algorithmic frameworks that can produce high-quality solutions with minimal computation time, which our approach aims to address. We do not aim to outperform highly specialized solvers, like Concorde and LKH-3. We have added language in the introduction and Section 2 (Related Work), paragraph 2, to update our statement.
>
> We have added experimental results for a variety of classical baselines, including Concorde, LKH-3, and greedy construction heuristics (Nearest Neighbor, Farthest Insertion) in Table 1 and Figure 3 (Appendix B). We have also evaluated the performance of our approach and the other learning based approaches when generalizing from the training distribution (consisting of randomly generated TSPs) to TSPLIB instances. These results are shown in Table 3 (Appendix C). We evaluate our approach and other learning based approaches using models trained on random 100-node TSPs and evaluated on TSPLIB instances with 50 to 200 nodes and 2D Euclidean distances (29 instances). We report mean computation time and optimality gap averaged over 10 runs per instance.
>
> Our motivation for using ML is that our measure of global regret is potentially an excellent guide for GLS, but is computationally intractable. Therefore, we use a GNN to produce a fast approximation of this value.
>
> Finally, we have referenced some additional recent approaches for solving the TSP. We have added to Section 2 (Related Work) to highlight the differences between these methods and our work, and to Section 6 (Discussion) in the discussion regarding symmetry.

---

> ### Author Response · Authors · 2021-11-26
> **Response to reviewer feedback**
>
> We thank the reviewer again for the detailed review. We believe we have addressed all the concerns that were raised (as described in our detailed rebuttal). In particular, we contextualized our work with respect to classical approaches, have added classical baselines, and evaluated our method on the TSPLIB benchmark problem set. Our approach dominates other ML-based approaches (it is faster and produces better solutions) when solving the uniform random problem sets and when solving the TSPLIB instances (see Tables 1 and 3). Please let us know if you need any further clarification.
>
> Average performance across 29 TSPLIB instances:
>
> | Metric                | Kool et al.   | Joshi et al.  | da Costa et al. | Ours         |
> |-----------------------|---------------|---------------|-----------------|--------------|
> | Mean computation time | **0.532±0.300**   | 7.000±2.415   | 31.766±4.543    | **10.420±0.630** |
> | Mean optimality gap   | **16.767±67.953** | 40.025±15.702 | 1.725±3.767     | **1.529±1.328**  |
> |                       |               |               |                 |              |
>
> Kool’s and our methods are Pareto optimal, and the others are totally dominated.

---

### Official Review · Reviewer_aECP · 2021-11-03

**Correctness:** 4
**Technical Novelty And Significance:** 3
**Empirical Novelty And Significance:** 3
**Recommendation:** 8
**Confidence:** 3

**Main Review:**

The paper is well-written and presents an interesting approach to solving TSP that can outperform some other existing approaches. The structure is appropriate, there is a very good review of related works, good description of the method, experiments, and results. There are also extensive supplementary materials. The introduced method is novel, might be significant, and the quality of this article seems to be on-par with other papers applying ML techniques to solve TSP published at top-tier conferences (which are also cited in this paper). The only weakness I see is the way of presenting the results in Fig. 3. All the percentages of optimally solved problems are relatively low, so the plots for some algorithms are not clearly visible (however, it is clear that the introduced algorithm outperforms other approaches). I recommend acceptance of this article.

**Summary Of The Paper:**

The paper presents a hybrid data-driven approach for solving the TSP based on Graph Neural Networks and Guided Local Search. The model predicts the regret of including each edge of the problem graph in the solution; GLS uses these predictions in conjunction with the original problem graph to find solutions. The experiments demonstrate that this approach converges to optimal solutions at a faster rate than state-of-the-art learning-based approaches and non-learning GLS algorithms for the TSP, finding optimal solutions to 96% of the 50-node problem set, 7% more than the next best benchmark, and to 20% of the 100-node problem set, 4.5× more than the next best benchmark. When generalizing from 20-node problems to the 100-node problem set, this approach finds solutions with an average optimality gap of 2.5%, a 10× improvement over the next best learning-based benchmark.

**Summary Of The Review:**

The paper is well-written and presents an interesting approach to solving TSP that can outperform some other existing approaches. The structure is appropriate, there is a very good review of related works, good description of the method, experiments, and results. There are also extensive supplementary materials. The introduced method is novel, might be significant, and the quality of this article seems to be on-par with other papers applying ML techniques to solve TSP published at top-tier conferences (which are also cited in this paper). The only weakness I see is the way of presenting the results in Fig. 3. All the percentages of optimally solved problems are relatively low, so the plots for some algorithms are not clearly visible (however, it is clear that the introduced algorithm outperforms other approaches). I recommend acceptance of this article.

---

> ### Author Response · Authors · 2021-11-23
> **Response to reviewer feedback**
>
> We thank the reviewer for their positive comments. We have adjusted the evaluation method to align better with recent computational guidelines for learning based approaches for the TSP. Specifically, we have added additional experiments where both solution quality and computation time are unfixed, shown in Table 1 and Figure 3 (Appendix B). These allow us to evaluate other approaches in a way that is closer to their original implementation. We have re-run our experiments for several approaches (Kool et al., Joshi et al.) using their original implementations, which resulted in performance improvements for these approaches. We have also added classical baselines to the results in Table 1, including well known and highly effective classical algorithms such as Concorde and LKH-3, as well as greedy heuristics (Nearest Neighbor, Farthest Insertion). The updated results are shown in Table 1 and 2, as well as Figures 3, 4 (Appendix B), and 5 (Appendix C).
>
> Furthermore, we have evaluated the performance of our approach and the other learning based approaches when generalizing from the training distribution (consisting of randomly generated TSPs) to TSPLIB instances, which are meant to represent real-world problem instances. These results are shown in Table 3 (Appendix C). We evaluate our approach and other learning based approaches using models trained on random 100-node TSPs and evaluated on TSPLIB instances with 50 to 200 nodes and 2D Euclidean distances (29 instances).
>
> We have also added to the introduction and Section 2 (Related Work) to strengthen our motivation for this work. We acknowledge that Concorde and LKH-3 can easily solve 100-node TSPs. However, we argue that Concorde and other highly-specialized solvers are not used to solve real-time routing problems in industry because practical problems often impose constraints on top of the basic TSP or CVRP problem definitions. Highly specialized solvers are difficult to adapt to new constraints. Thus, there is a need for general algorithmic frameworks that can produce high-quality solutions with minimal computation time, which our approach aims to address. Such general frameworks can be easily adapted to various problem definitions with additional constraints while remaining performant.
>
> Finally, we have referenced some additional recent approaches for solving the TSP. We have added to Section 2 (Related Work) to highlight the differences between these methods and our work, and to Section 6 (Discussion) in the discussion regarding symmetry.

---

### Author Response · Authors · 2021-11-23
**Response to reviewer feedback**

We thank the reviewers for their comments and suggestions.

We have adjusted the evaluation method to align better with recent computational guidelines for learning based approaches for the TSP. Specifically, we have added additional experiments where both solution quality and computation time are unfixed, shown in Table 1 and Figure 3 (Appendix B). These allow us to evaluate other approaches in a way that is closer to their original implementation. We have re-run our experiments for several approaches (Kool et al., Joshi et al.) using their original implementations, which resulted in performance improvements for these approaches. We have also added classical baselines to the results in Table 1, including well known and highly effective classical algorithms such as Concorde and LKH-3, as well as greedy heuristics (Nearest Neighbor, Farthest Insertion). The updated results are shown in Table 1 and 2, as well as Figures 3, 4 (Appendix B), and 5 (Appendix C).

Furthermore, we have evaluated the performance of our approach and the other learning based approaches when generalizing from the training distribution (consisting of randomly generated TSPs) to TSPLIB instances, which are meant to represent real-world problem instances. These results are shown in Table 3 (Appendix C). We evaluate our approach and other learning based approaches using models trained on random 100-node TSPs and evaluated on TSPLIB instances with 50 to 200 nodes and 2D Euclidean distances (29 instances).

We have also added to the introduction and Section 2 (Related Work) to strengthen our motivation for this work. We acknowledge that Concorde and LKH-3 can easily solve 100-node TSPs. However, we argue that Concorde and other highly-specialized solvers are not used to solve real-time routing problems in industry because practical problems often impose constraints on top of the basic TSP or CVRP problem definitions. Highly specialized solvers are difficult to adapt to new constraints. Thus, there is a need for general algorithmic frameworks that can produce high-quality solutions with minimal computation time, which our approach aims to address. Such general frameworks can be easily adapted to various problem definitions with additional constraints while remaining performant.

Finally, we have referenced some additional recent approaches for solving the TSP. We have added [1, 2] to Section 2 (Related Work) and highlighted the differences between these methods and our work. These approaches construct tours using heuristics, whereas we are the first to use a metaheuristic. We have added [3] to Section 2, and referenced it in Section 6 (Discussion) in the discussion regarding symmetry.

[1] Wouter Kool, Herke van Hoof, Joaquim Gromicho, Max Welling. Deep Policy Dynamic Programming for Vehicle Routing Problems. arXiv preprint, 2021.

[2] Zhang-Hua Fu, Kai-Bin Qiu, Hongyuan Zha. Generalize a small pretrained model to arbitrarily large tsp instances. AAAI 2021.

[3] Yeong-Dae Kwon, Jinho Choo, Byoungjip Kim, Iljoo Yoon, Youngjune Gwon, Seungjai Min. POMO: Policy Optimization with Multiple Optima for Reinforcement Learning. NeurIPS 2020.

---

### Decision · Program_Chairs · 2022-01-20

**Decision:**

Accept (Poster)

**Comment:**

In this paper, a novel machine learning-based method for solving TSP is presented; this method uses guided local search in conjunction with a graph neural network, which is trained to predict regret. Reviewers disagree rather sharply on the merits of the paper. Three reviewers think that the paper is novel, interesting, and has good empirical results. Two reviewers think that the fact the results are not competitive with the best non-learning-based ("classic") solvers mean that the paper should be rejected.
This area chair believes that research is fundamentally not about beating benchmarks, but about new, interesting, and sound ideas. The conceptual novelty of this method, together with the good results compared with other learning-based methods, is sufficient for accepting the paper.